# Phase separation of the PRPP amidotransferase into dynamic condensates promotes de novo purine synthesis in yeast

Masak Takaine[1,2*], Rikuri Morita[3], Yuto Yoshinari[4], Takashi Nishimura[4]

1 GIAR, Gunma University, Maebashi, Japan, 2 IMCR, Gunma University, Maebashi, Japan, 3 Center for Computational Sciences, University of Tsukuba, Ibaraki, Japan, 4 Laboratory of Metabolic Regulation and Genetics, Institute for Molecular and Cellular Regulation, Gunma University, Maebashi, Gunma, Japan

* masaktakaine@gmail.com

## Abstract

De novo purine synthesis (DPS) is up-regulated under conditions of high purine demand to ensure the production of genetic materials and chemical energy, thereby supporting cell proliferation. However, the regulatory mechanisms governing DPS remain unclear. We herein show that PRPP amidotransferase (PPAT), the rate-limiting enzyme in DPS, forms dynamic and motile condensates in *Saccharomyces cerevisiae* cells under a purine-depleted environment. The formation and maintenance of condensates requires phase separation, which is driven by target of rapamycin complex 1 (TORC1)-induced ribosome biosynthesis. The self-assembly of PPAT molecules facilitates condensate formation, with intracellular PRPP and purine nucleotides both regulating this self-assembly. Moreover, molecular dynamics simulations suggest that clustering-mediated PPAT activation occurs through intermolecular substrate channeling. Cells unable to form PPAT condensates exhibit growth defects, highlighting the physiological importance of condensation. These results indicate that PPAT condensation is an adaptive mechanism that regulates DPS in response to both TORC1 activity and cellular purine demands.

## Introduction

Purine nucleotides and their derivatives play crucial roles in many cellular processes, making them essential metabolites for all living organisms. The purine nucleotide triphosphates, ATP and GTP, act as cellular energy currencies. In addition, purine metabolites are integrated into genetic materials, such as DNA and RNA, and into coenzymes, including NAD and coenzyme A. Purine nucleotides are synthesized through salvage or de novo pathways. Under normal conditions, cells primarily synthesize the majority of purine nucleotides via the salvage pathway by adding phosphoribosyl pyrophosphate (PRPP), derived from the pentose phosphate pathway, to purine bases, which are often imported from outside the cell. Since at least one of the

**Data availability statement:** All relevant data are within the paper and its Supporting Information files.

**Funding:** This work was supported by the Gout and Uric Acid Foundation of Japan (https://www.tufu.or.jp, 2019 Research Grants No. 8 to MT), the Japan Society for the Promotion of Science (https://www.jsps.go.jp, KAKENHI Grant Number JP22K06216 to MT), and the Institute for Molecular and Cellular Regulation, Gunma University (https://www.imcr.gunma-u.ac.jp, The Joint Research Program 2022, No. 21005 to RM and MT). The funders had no role in study design, data collection and analysis, decision to publish, or preparation of the manuscript.

**Competing interests:** The authors have declared that no competing interests exist.

**Abbreviations:** AMP, adenosine 5´-monophosphate; ANOVA, analysis of variance; AS, ammonium sulfate; DPS, de novo purine synthesis; EB, extraction buffer; FC, fold change; FDR, false discovery rate; GMP, guanine 5´-monophosphate; HGPRT, hypoxanthine-guanine phosphoribosyltransferase; IDR, intrinsically disordered region; IMP, inosine 5´-monophosphate; MD, molecular dynamics; PEG, polyethylene glycol; PLS-DA, partial least squares discriminant analysis; SAH, S-adenosylhomocysteine; SD, standard deviation.

two pathways is required for purine synthesis, they provide a fail-safe mechanism for each other.

The de novo purine biosynthetic pathway consists of 10 highly conserved sequential chemical reactions facilitated by six enzymes that convert PRPP to inosine 5´-monophosphate (IMP). IMP is then converted into adenosine 5´-monophosphate (AMP) or guanine 5´-monophosphate (GMP) through two branched paths. Among these enzymes, PRPP amidotransferase (PPAT) (EC 2.4.2.14) is the rate-limiting enzyme and plays a pivotal role in the regulation of de novo purine synthesis (DPS). PPAT catalyzes the initial step of the de novo pathway, namely, the transfer of amino groups from glutamine to PRPP to produce 5-phosphoribosylamine (PRA) (Eq 1). This reaction is further divided into two half-reactions: the release of ammonia ($NH_3$) from glutamine (Eq 2) and the ligation of $NH_3$ with PRPP (Eq 3). PPAT is a bifunctional enzyme with two catalytic domains, possessing an N-terminal glutamine amidotransferase (GATase) domain that catalyzes the former reaction, and a C-terminal phosphoribosyltransferase (PRTase) domain for the latter.

$$PRPP + glutamine + H_2O \rightarrow PRA + glutamate + PP_i \qquad (1)$$

$$glutamine + H_2O \rightarrow NH_3 + glutamate \qquad (2)$$

$$PRPP + NH_3 \rightarrow PRA + PP_i. \qquad (3)$$

While the biochemical properties of each enzyme involved in purine synthesis have been extensively examined, the regulation of DPS within cells remains unclear. In bacterial and mammalian cells, when extracellular purine bases are available, purine nucleotides are preferentially synthesized by recycling bases through the salvage pathway, while the de novo pathway is strongly inhibited [1,2]. Conversely, under conditions of purine deprivation or high purine demand (*e.g.*, in actively proliferating cells), the de novo pathway is activated to provide sufficient levels of purine nucleotides [3]. Although the mechanisms underlying these regulatory processes have not yet been elucidated in detail, cells appear to limit the use of the de novo pathway in order to conserve energy and resources because the generation of each IMP requires up to five ATP and four amino acids.

The enzyme activity of PPAT is subject to feedback inhibition by the end products of the de novo pathway, such as AMP and GMP [4–6]. This inhibition is considered to play a role in both suppressing DPS in the presence of salvageable extracellular purine bases and activating it during purine starvation [4]. However, millimolar levels of AMP or GMP are required to inhibit PPAT catalytic activity [4,6,7], whereas the intracellular concentrations of these nucleotides are markedly lower [8], raising questions about the physiological relevance of this feedback inhibition. Other regulatory layers of PPAT activity, such as post-translational modifications and intracellular signaling, may also play a role but have yet to be investigated.

Emerging evidence suggests that many metabolic enzymes, including PPAT, form condensates in response to external stimuli in order to regulate cellular metabolism [9]. For example, the pyruvate kinase Cdc19 in budding yeast reversibly aggregates into large foci upon glucose starvation and heat shock [10,11]. This aggregation protects Cdc19 from stress-induced degradation and inactivates it, both of which are crucial for the efficient formation of stress granules and cell regrowth after stress. Similarly, PPAT reversibly localizes to cytoplasmic condensates upon the depletion of extracellular purine bases in budding yeast and HeLa cells [12,13]. Recent studies indicated that in HeLa cells, PPAT also colocalized with other enzymes involved in DPS within a multi-enzyme cellular complex known as a purinosome, which appeared to enhance pathway flux near mitochondria [14]. However, the mechanisms regulating the formation of PPAT condensates and its physiological significance remain unclear.

We herein demonstrate that PPAT forms dynamic intracellular condensates through phase separation in budding yeast cells in the absence of external purine bases. Active ribosome synthesis promoted by target of rapamycin complex 1 (TORC1) is crucial for the molecular crowding that drives phase transition. An in vitro analysis showed that the self-assembly of PPAT alone is sufficient for condensate formation, which is inhibited by purine nucleotides. Moreover, increased intracellular PRPP promotes the condensation of PPAT, opposing the inhibitory effects of purine nucleotides. Molecular dynamics (MD) simulations suggest the condensation-dependent activation of the enzyme by intermolecular $NH_3$ channeling. Consistent with these results, PPAT condensates play a vital role in DPS to ensure cell proliferation. Collectively, we propose that PPAT condensation by phase transition is a rapid regulatory mechanism for DPS in response to both TORC1 activity and available purine bases.

## Results

### The yeast PPAT Ade4 concentrates in fine particles

Previous semi-comprehensive analyses revealed that the budding yeast PPAT Ade4 formed cytoplasmic foci in response to environmental purine bases [12,15]. However, the dynamics of the assembly and disassembly of Ade4 foci, their physical properties, and the mechanisms underlying the formation of foci remain unclear. Therefore, we reexamined Ade4 foci using a cell biological approach. In the present study, adenine was used as a representative purine base because of its high solubility in medium and its role as a precursor for both AMP and GMP through reverse conversion from AMP to IMP (Fig 1A). We visualized the localization of the Ade4 protein by tagging the chromosomal *ADE4* gene with a yeast codon-optimized GFP (hereafter referred to as GFP). As demonstrated in previous studies [12,15], Ade4-GFP showed a large focus in cells transferred into medium lacking purine bases, but not in the presence of adenine (Fig 1B). Unexpectedly, Ade4 tagged with a GFP bearing the monomeric mutation A206K [16] (Ade4-mGFP) appeared as numerous fine dim dots under the same conditions (Fig 1B). Consistently, Ade4 tagged with monomeric NeonGreen (Ade4-mNG) formed fine particles with markedly higher fluorescence intensities (Fig 1B and 1C). Ade4 tagged with the monomeric red fluorescent protein mKate2 (Ade4-mKate2) showed similar particle localization (S1A Fig). Therefore, we conclude that Ade4-GFP concentrates into 1–2 foci due to GFP dimerization.

Z-stack imaging revealed that Ade4-mNG particles localized throughout the cytoplasm (Fig 1D). Time-lapse imaging further demonstrated that these particles were dynamic structures: they assembled within 40 min after the depletion of extracellular adenine (Figs 1E, 1G, S1B, and S1–S3 Movies) and disassembled within 10 min after adenine supplementation (Figs 1F, 1H, S1C, and S4–S6 Movies). Closer time-lapse imaging indicated the high motility of particles: they moved freely in three dimensions and repeatedly moved in and out of the focal plane within seconds (Figs 1I–1K, S1D–S1F, and S7 Movie). These results suggest that Ade4 forms dynamic and motile particles in response to the availability of purines.

In subsequent experiments, we primarily focused on Ade4-mNG particles to elucidate the molecular mechanisms regulating Ade4 condensation. We also examined the assembly of Ade4-GFP foci due to their similar properties to Ade4-mNG particles and their suitability for high-content imaging because of their higher brightness.

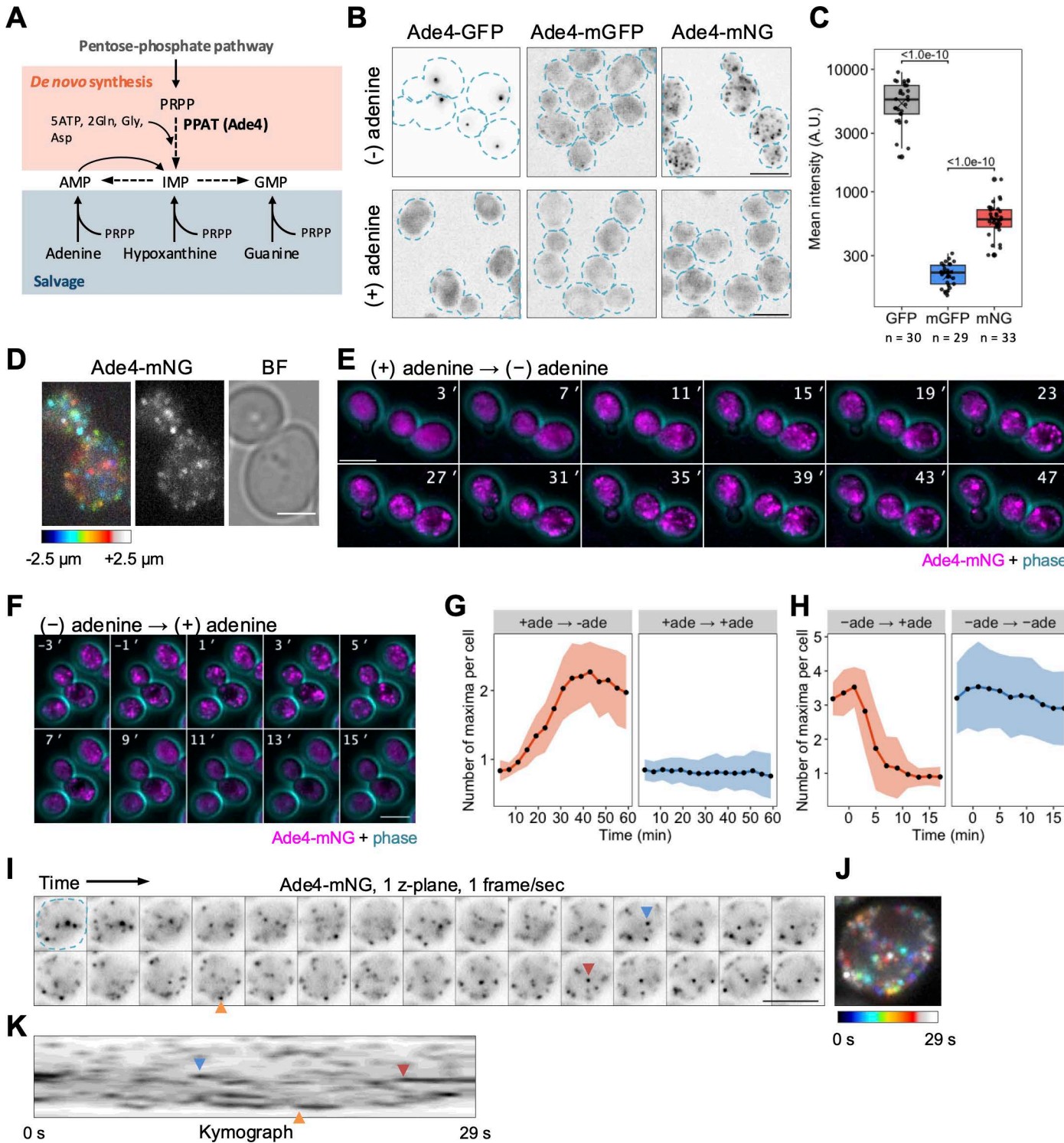

**Fig 1. The yeast PPAT Ade4 concentrates in fine particles.**
(A) Schematic of purine de novo and salvage pathways. (B) Fluorescent images of cells expressing Ade4 tagged C-terminally with the indicated green fluorescent protein. Cells were grown in medium containing 0.02 mg/mL adenine, washed with medium with or without adenine, and then incubated in the same medium for 45 min before imaging. The intensity of GFP is represented by an inverted grayscale. Dotted lines indicate the cell outline. (C) Quantification of the mean fluorescence intensity of Ade4 condensates observed in cells as shown in (B). Data are represented by box and dot plots.

Cross marks indicate the population mean. *P*-values vs. mGFP are shown (Steel's multiple comparison test). (D) A higher resolution image of Ade4-mNG particles. Z-positions are represented by a color code (left). A grayscale image of particles (middle) and a bright field image of cells (right) are also shown. (E) Time-lapse imaging of the assembly of Ade4-mNG particles by adenine depletion. Cells were grown in medium containing 0.02-mg/mL adenine and washed with medium lacking adenine at *t* = 0 min. (F) Time-lapse imaging of the disassembly of Ade4-mNG particles by adenine supplementation. Cells were grown in the absence of adenine and 0.02-mg/mL adenine was supplemented at *t* = 0 min. **(G,** H) Panels **G** and **H** show the quantification of data in **(E)** and (F), respectively. The number of intracellular Ade4-mNG particles was counted as the number of maxima of green fluorescence intensity per cell. Data are the mean of 10–12 independent fields of view, and ~70 cells were examined per field. The shaded area indicates ± 1 SD. **(I**–K) Closer time-lapse imaging of Ade4-mNG particles. A cell bearing the particles (outlined by a dotted line in the first frame) was imaged on a single z-plane at 1-s intervals. The intensity of mNG is represented by an inverted grayscale. Panel **J** shows a color-coded superimposition of the images from each frame shown in (I). Panel **K** is a kymograph of the images shown in (I). Arrowheads indicate the appearance of some particles. Scale bars = 5 μm. The data for panels **C**, **G**, and **H** may be found in S1 Data.

## Ade4 particles possess liquid-like properties

The reversible assembly and dynamic nature of Ade4-mNG particles prompted us to hypothesize that they are liquid-like assemblies rather than static, solid-like aggregates. Therefore, we treated cells containing Ade4-mNG with 1,6-hexanediol, an aliphatic alcohol that dissolves various intracellular membrane-less structures [17]. The permeabilization of yeast cells with digitonin did not affect the assembly of Ade4-mNG particles, whereas particles exposed to 10% 1,6-hexanediol plus digitonin significantly disassembled within 10 min, confirming their liquid-like properties (Fig 2A and 2B).

Previous studies highlighted the role of an intrinsically disordered region (IDR) in the formation of intracellular membrane-less structures [18]. We analyzed the amino acid sequences of Ade4 and other PPATs using the DISOPRED algorithm [19], a neural network-based method that predicts IDRs. We identified an IDR at the C terminus of Ade4 (Fig 2C). The presence of the IDR was highly conserved across Ade4 homologues in Ascomycota and all fungi (S2A Fig), as well as in PPATs from other kingdoms, including bacteria and mammals (S2B Fig). However, the amino acid sequences of IDRs varied across species, even among yeasts (S2C and S2D Fig). An Ade4 mutant lacking the IDR, Ade4ΔIDR-mNG, was incapable of forming punctate structures, suggesting the necessity of the IDR for Ade4 particle assembly (Fig 2D). Collectively, these results suggest that Ade4 molecules condensate into fine particles through IDR-dependent phase separation.

## Formation of Ade4 condensates requires active TORC1 and ribosome biogenesis

We examined the factors required for phase separation, which facilitates the assembly of Ade4 condensates. Recent studies suggested that TORC1 controlled phase separation in the cytoplasm of both yeast and mammalian cells by modulating ribosome crowding [20]. Therefore, we investigated the role of TORC1 in the formation of Ade4 condensates. In cells, rapamycin binds to FKBP12, forming a complex that effectively inhibits TORC1 [21]. The addition of rapamycin suppressed the assembly of Ade4-GFP foci in wild-type cells, but not in cells lacking Fpr1 (budding yeast FKBP1) (Fig 2E). Similarly, rapamycin significantly inhibited the assembly of Ade4-mNG condensates in wild-type cells (Fig 2F and 2G). These results indicate that TORC1 activity is essential for the formation of Ade4 condensates.

To further delineate the TORC1-related factors responsible for Ade4 condensate formation, we examined Ade4-GFP foci in 37 single-gene deletion mutants, each lacking one of the 17 upstream regulators, the 17 downstream effectors of TORC1 signaling, or 3 genes of interest (S3A and S3B Fig). Consistent with the inhibitory effects of rapamycin, the assembly of Ade4-GFP foci was typically reduced in mutants of the upstream activators of TORC1 signaling (such as the Pib2 or EGO complex). In contrast, most of the downstream effectors of TORC1 signaling exerted negligible or no effects on the formation of Ade4-GFP foci. However, the formation of Ade4-GFP foci and Ade4-mNG condensates was significantly suppressed in *sfp1Δ* and *sit4Δ* cells (Figs 2H, 2I, S3A, and S3B).

Sfp1 is a transcriptional activator that regulates the expression of ribosomal protein and ribosome biogenesis genes [22]. Sit4 is a subunit of type 2A protein phosphatase, the activity of which is controlled by TORC1 [23]. Both have been implicated in cytoplasmic crowding through ribosome biogenesis downstream of TORC1 [20,24,25]. Therefore,

PLOS Biology

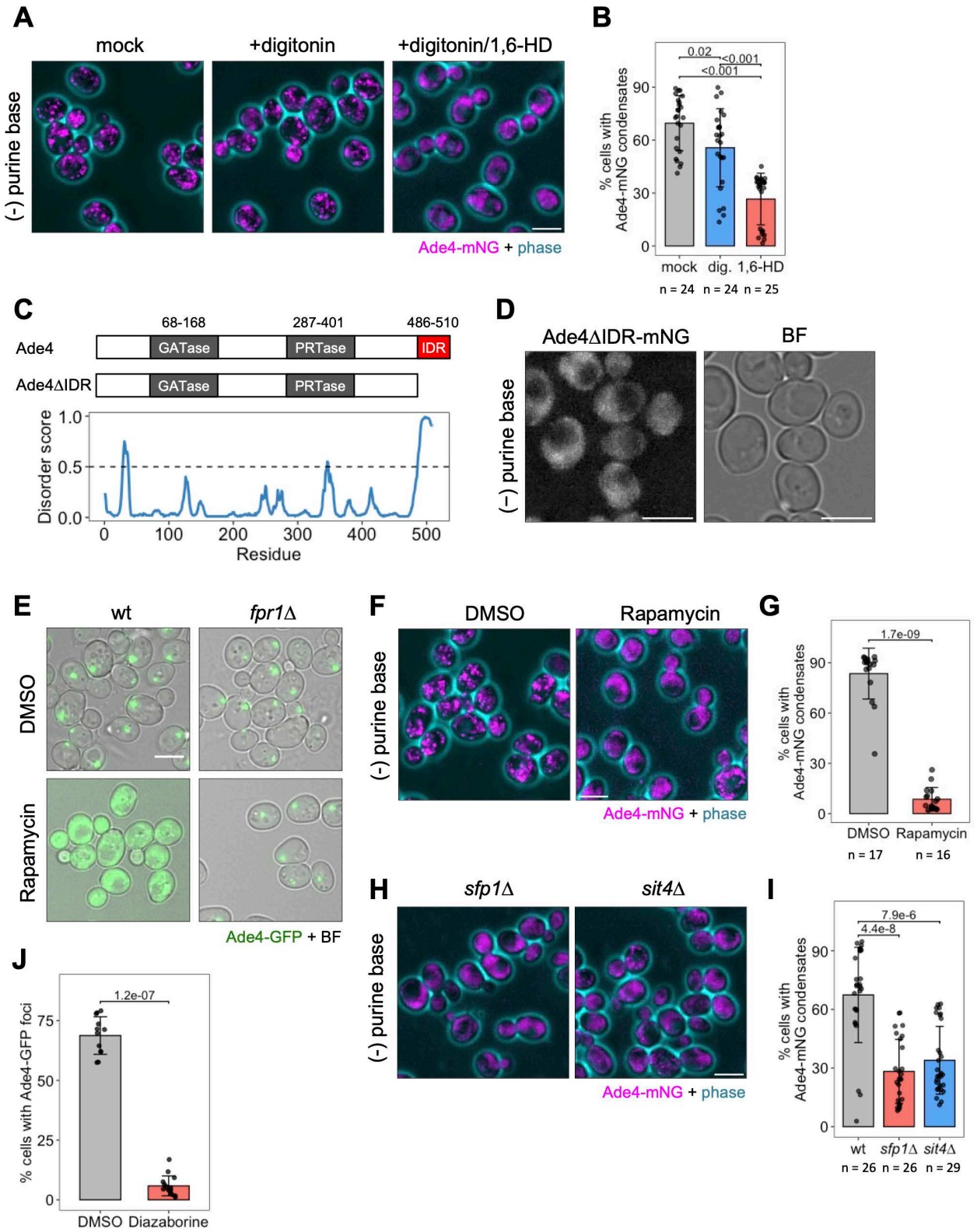

**Fig 2. Ade4 condensate formation requires active TORC1 and ribosome biogenesis.** (A) The aliphatic alcohol 1,6-hexanediol dissolves Ade4 condensates. Wild-type cells harboring Ade4-mNG condensates were treated with 2-μg/mL digitonin ± 10% (w/v) 1,6-hexanediol for 10 min and then

imaged. (B) Quantification of data shown in (A). The percentage of cells with Ade4-mNG condensates per field of view (containing 361 ± 159 cells) was plotted. Data were pooled from three independent experiments. Bars and error bars indicate the mean ± 1 SD. *P*-values were calculated using the two-tailed Steel–Dwass multiple comparison test. (C) Domain structures of the budding yeast Ade4 and the IDR mutant. The bottom graph shows disorder predictions. (D) The C-terminal IDR of Ade4 is required for condensate formation. Cells expressing the mutant Ade4 lacking the C-terminal IDR tagged with mNG were incubated in the absence of purine bases for 45 min and imaged. (E) Wild-type and *fpr1Δ* cells expressing Ade4-GFP were grown in medium without adenine in the presence of 0.02% (v/v) DMSO or 0.02% DMSO plus 2-µg/mL rapamycin for 45 min before imaging. (F) Rapamycin suppresses the assembly of Ade4-mNG condensates. Wild-type cells expressing Ade4-mNG were treated with rapamycin and imaged as described in (E). (G) Quantification of data shown in (F). The percentage of cells with Ade4-mNG condensates per field of view (containing 443 ± 138 cells) was plotted. Data were pooled from two independent experiments. Bars and error bars indicate the mean ± 1 SD. *P*-values were calculated using the two-tailed Mann–Whitney U test. (H) The assembly of Ade4-mNG condensates was significantly suppressed in *sfp1Δ* and *sit4Δ* cells. (I) Quantification of data shown in (H). The percentage of cells with Ade4-mNG condensates per field of view (containing 517 ± 123 cells) was plotted. Data were pooled from three independent experiments. Bars and error bars indicate the mean ± 1 SD. *P*-values vs. wt were calculated using the two-tailed Steel's multiple comparison test. (J) The inhibition of ribosome synthesis suppresses the assembly of Ade4 condensates. Cells expressing Ade4-GFP were grown in medium containing adenine and then incubated in medium without adenine in the presence of 0.05% (v/v) DMSO or 0.05% DMSO plus 5-µg/mL diazaborine for 45 min before imaging. The percentage of cells with Ade4-GFP foci per field of view (containing 780 ± 214 cells) was plotted. Data were pooled from two independent experiments. Bars and error bars indicate the mean ± 1SD. *P*-values were calculated using the two-tailed Mann–Whitney U test. Representative cell images are shown in S3C Fig. Scale bars = 5 µm. The data for panels **B**, **C**, **G**, **I**, and **J** may be found in S1 Data.

we evaluated the importance of ribosome synthesis in Ade4 condensate formation. Diazaborine specifically inhibits the maturation of the 60S ribosomal subunit and reduces the level of the 60S subunit in yeast cells [26]. The treatment with diazaborine strongly suppressed the formation of Ade4-GFP foci and Ade4-mNG condensates, indicating that the active synthesis of the ribosomal 60S subunit was required for the assembly of these condensates (Figs 2J, S3C, and S3D). Collectively, these results suggest that macromolecular crowding of the cytoplasm, driven by TORC1-mediated ribosome synthesis, is a primary mechanism facilitating the phase separation required for Ade4 condensation.

We also examined the contribution of TORC1 and ribosome synthesis to the maintenance of Ade4 condensation. The treatment with rapamycin of cells with preformed Ade4-GFP foci rapidly disassembled foci within 30 min (S4 Fig). The treatment with diazaborine resulted in the disassembly of foci, similar to rapamycin, but at a slower disassembly rate (S4 Fig). These results suggest that the activity of TORC1 and ribosome synthesis are required not only for the assembly of Ade4 condensates, but also for their maintenance.

### Ade4 protein condensates into particles under molecular crowding conditions in vitro

Two hypotheses have been proposed for the molecular mechanisms regulating the assembly of Ade4 condensates. The first hypothesis suggests that Ade4 alone is sufficient for condensate formation, while the second posits that additional cellular components, such as proteins, lipids, and organelles, are required for Ade4 to condensate into particles. To distinguish between these possibilities, we investigated the properties of the purified Ade4 protein in vitro. Previous attempts to express Ade4 in its active form in bacteria were unsuccessful [27], prompting us to purify Ade4-mNG directly from budding yeast cells (S5A Fig). Based on the findings of 20 quantitative proteomic studies, the intracellular concentrations of Ade4 were estimated to be ~0.5 µM (S6 Fig). In our in vitro assays, we selected concentrations of Ade4-mNG that were close to or slightly lower than this value.

Under physiological ionic conditions (150-mM NaCl and 4-mM $MgCl_2$, pH 7.5), purified Ade4-mNG dispersed homogenously, showing no signs of condensate formation ("0%" in Fig 3A). We then examined the effects of macromolecular crowding using polyethylene glycol (PEG) as a crowding agent. The addition of PEG8000 (PEG8K) led to the formation of sub-micrometer-sized particles of Ade4-mNG in a concentration-dependent manner; significant amounts of particles formed at concentrations higher than 10% PEG (Fig 3A–3C). Controls with the mNG protein alone did not form particles in 10% PEG (Fig 3D). We also investigated the effects of different sizes of PEG and found that Ade4-mNG self-assembled into particles in the presence of PEG with molecular weights higher than 4 kDa (S5B–S5D Fig). Consistent with this size

PLOS Biology

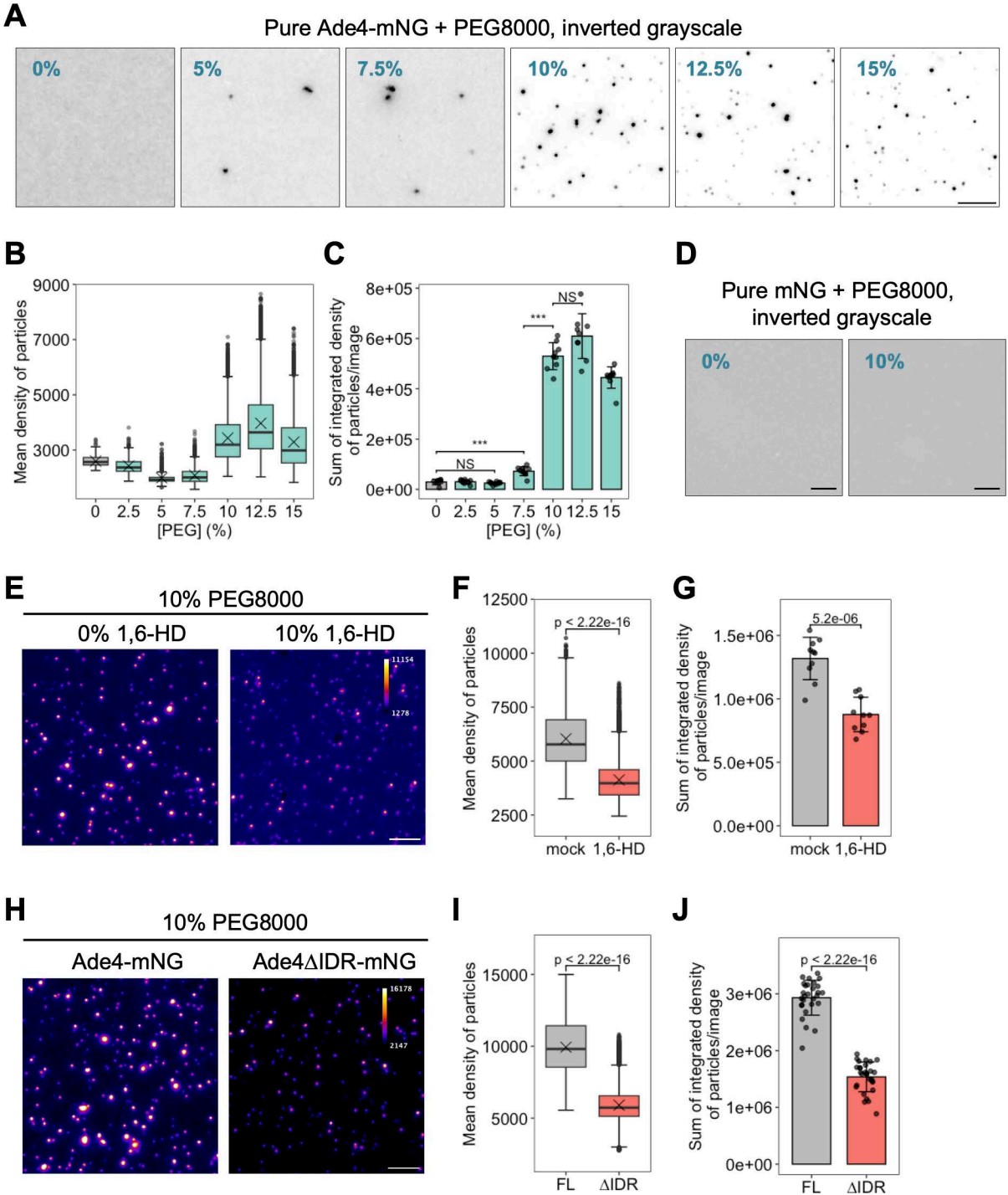

**Fig 3. The Ade4 protein condensates into particles under molecular crowding conditions in vitro.** (A) Purified Ade4-mNG condensated into fine particles in a PEG concentration-dependent manner. Ade4-mNG (0.15 μM) was supplemented with PEG8K at the indicated concentration (%, w/v). Inverted grayscale images are shown. (B, C) Quantification of data shown in (A). The mean fluorescence intensities of Ade4-mNG particles were box plotted in (B). Cross marks indicate the population mean. The integrated fluorescence intensities of particles were summed per field and plotted in (C). Bars and error bars indicate the mean ± 1SD. Seven to 10 fields of view were imaged for each condition and an average of 390 particles per field were examined. *P*-values were calculated using the two-tailed Steel–Dwass multiple comparison test. (D) The purified mNG protein did not condensate into particles even under molecular crowding conditions. The supplementation of 0.14-μM mNG with 0 or 10% PEG8K was performed. (E) 1,6-Hexanediol

attenuated the in vitro condensation of Ade4-mNG. Ade4-mNG (0.15 µM) was supplemented with 10% PEG8K in the presence of 0 or 10% (w/v) 1,6-hexanediol. Fluorescence intensity is represented by pseudo-colors. **(F, G)** Quantification of data shown in (E). The mean fluorescence intensities of particles were box plotted in (F). Cross marks indicate the population mean. The integrated fluorescence intensities of particles were summed per field and plotted in (G). Bars and error bars indicate the mean ± 1 SD. Ten fields of view were imaged for each condition, and an average of 1,197 particles per field were examined. $P$-values were calculated using the two-tailed Mann–Whitney U test **(F)** or Welch's $t$ test (G). (H) Ade4ΔIDR-mNG condensated into particles less effectively than Ade4-mNG in vitro. The supplementation of 0.2-µM Ade4-mNG or Ade4ΔIDR-mNG with 10% PEG8K was performed. Fluorescence intensity is represented by pseudo-colors. **(I,** J) Quantification of data shown in (H). The mean fluorescence intensities of particles were box plotted in (I). Cross marks indicate the population mean. The integrated fluorescence intensities of particles were summed per field and plotted in (J). Bars and error bars indicate the mean ± 1 SD. Twenty-eight fields of view were imaged for each condition, and an average of 1,341 particles per field were examined. $P$-values were calculated using the two-tailed Mann–Whitney U test. Scale bars = 5 µm. The data for panels **B, C, F, G, I,** and **J** may be found in S1 Data.

dependency, other crowding agents, such as 2% lysozyme (MW: 14 kDa) and Ficoll-400 (MW: 400 kDa), also induced the assembly of Ade4-mNG particles (S5E Fig). These results suggest that Ade4 self-assembles into sub-micrometer particles that are similar in size to intracellular condensates under macromolecular crowding conditions.

We then investigated the role of phase separation in the self-assembly of Ade4-mNG in vitro. In the presence of 1,6-hexanediol, marked decreases were observed in the mean fluorescence intensity and total mass of Ade4-mNG particles (Fig 3E–3G). Consistently, the deletion of IDR from Ade4 markedly reduced condensate formation; however, the Ade4ΔIDR-mNG protein still formed visible particles (Fig 3H–3J). These results suggest that while Ade4 molecules form condensates independent of the C-terminal IDR moiety, phase separation is responsible for the complete assembly of Ade4 particles in vitro.

## Metabolomics analysis of budding yeast cells grown in the absence and presence of adenine

We investigated the mechanisms by which intracellular Ade4 condensates assemble in response to environmental purine deprivation. Nutrients such as amino acids and glucose typically regulate the activity of TORC1 [28]. Previous findings [29,30] suggest that environmental purine bases do not affect TORC1 activity in mammalian cells (although this was not the main focus of these studies). To investigate whether TORC1 activity in yeast is also independent of environmental purine bases, we examined the phosphorylation state of ribosomal protein S6 (Rps6), which is a reliable reporter of TORC1 activity [31]. The treatment of yeast cells with rapamycin for 30 min had no effect on the total amount of Rps6 but significantly reduced the intensity of the band of phosphorylated Rps6 (p-Rps6), which confirmed that the phosphorylation of Rps6 correlated with TORC1 activity (S7A Fig). In contrast to rapamycin, the treatment of cells cultured in the absence of purine bases with adenine or hypoxanthine for 30 min did not significantly affect the phosphorylation state of Rps6 (S7B Fig). The effect of removing purine bases from the medium on the phosphorylation state of Rps6 was also examined. Cells grown in the presence of adenine or hypoxanthine were washed and cultured in the absence of purine bases for an additional 40 min; however, the phosphorylation state of Rps6 remained largely unchanged (S7C Fig). These results suggest that the addition or removal of purine bases in the medium did not affect TORC1 activity in yeast cells, at least for short periods of time, namely, 30–40 min.

Based on these results, TORC1-mediated macromolecular crowding, which drives Ade4 condensate assembly, appears to be independent of extracellular purine base levels. Therefore, there may be additional layers of regulation that suppress or promote condensate formation based on the availability of environmental purines. We hypothesized that metabolites related to the incorporated purine bases regulate the assembly of Ade4 condensates.

To identify these functional metabolites, we conducted a comprehensive metabolomics analysis using liquid chromatography-tandem mass spectrometry. Wild-type cells were cultured in the absence or presence of adenine for 1, 7, and >24 h and were then collected for metabolic profiling (Fig 4A). A supervised classification method, a sparse partial least squares discriminant analysis (PLS-DA), demonstrated that samples were distinctly separated based on the

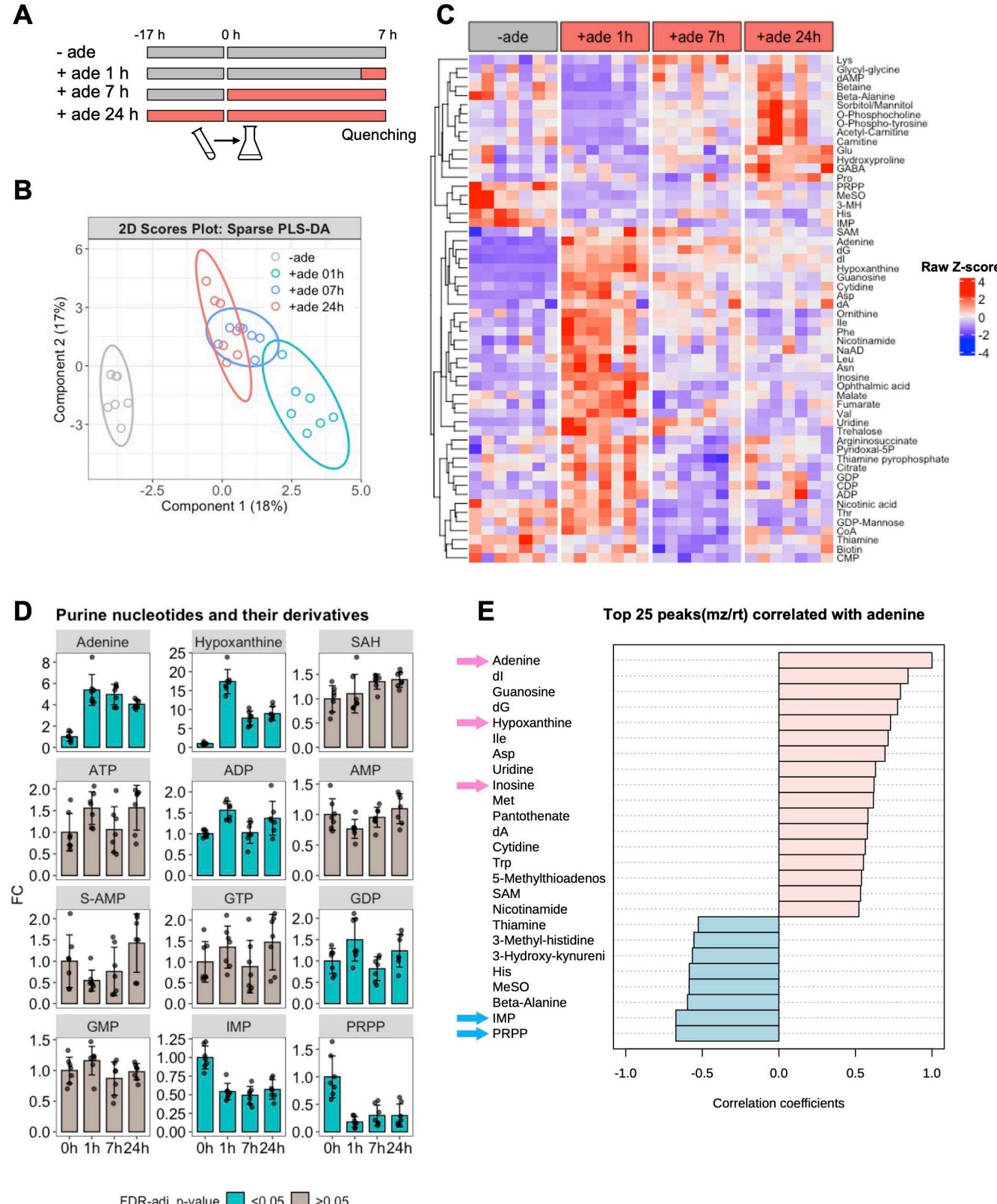

**Fig 4. Metabolomics analysis of budding yeast cells grown in the absence and presence of adenine.** (A) Experimental scheme for adenine exposure and cell quenching. The pre-culture was inoculated into the main culture at 0 h. (B) Sparse PLS-DA of metabolome samples. A 2D score plot of the

first two components with ellipse plots (95% confidence level). Data may be found in S1 Table. (C) Clustered heatmap of significantly changed metabolites. Rows indicate 56 metabolites that changed significantly between the indicated conditions (FDR-adjusted p-value < 0.05). Columns represent seven independent samples for each condition. Values for each metabolite were converted to Z-scores and plotted. Metabolites were clustered using the complete linkage method and Euclidean distance measure. Data may be found in S2 Table. (D) Changes in purine nucleotides and their derivatives. Data on the indicated metabolites were converted to fold change (FC) relative to the mean at 0 h (in the absence of adenine) and plotted. Bars and error bars indicate the mean ± 1 SD calculated from 7 independent biological replicates for each condition. The color of the bar indicates whether the change in the metabolite was significant. Data may be found in S3 Table. (E) Correlation coefficients of the top 25 metabolites correlated with intracellular adenine. Some metabolites of interest are indicated with arrows. The data for panel **E** may be found in S1 Data.

presence of environmental adenine and the duration of exposure to adenine (Fig 4B and S1 Table). A one-way analysis of variance (ANOVA) identified 56 metabolites whose concentrations were significantly affected by adenine (Fig 4C and S2 Table). The metabolic profile changed over the duration of adenine exposure, indicating both short- and long-term responses (Fig 4B and 4C). Intracellular concentrations of adenine increased more than 5-fold within 1 h of the addition of adenine (Fig 4D and S3 Table). A correlation analysis revealed that the concentrations of several metabolites, such as hypoxanthine, inosine, and deoxynucleotides, positively correlated with the change in intracellular adenine (Fig 4E). Among them, hypoxanthine, which is directly converted from adenine, showed the greatest increase after adenine supplementation and remained elevated (Fig 4D). Purine nucleotides and their derivatives slightly increased in the presence of adenine, with only some of these changes being significant (Figs 4D and S8A).

### Intracellular derivatives of purine nucleotides regulate the formation of Ade4 condensates

Since intracellular concentrations of adenine and hypoxanthine rapidly and significantly increased upon the addition of adenine, we speculated that these metabolites may suppress the formation of intracellular Ade4 condensates. However, these purine bases were converted to various metabolites after active uptake into cells through the purine-cytosine permease Fcy2 [32] (Fig 5A). Therefore, we used mutant strains to clarify whether adenine and hypoxanthine suppressed Ade4 condensate formation. When wild-type cells were treated with medium containing adenine or hypoxanthine, Ade4-GFP foci were completely disassembled (Fig 5B–5E, wt), indicating that hypoxanthine acted as an adenine substitute. In *aah1Δapt1Δ* cells, in which incorporated adenine was not metabolized further, the addition of adenine failed to disassemble Ade4 foci, whereas that of hypoxanthine completely disassembled foci (Fig 5B and 5C), suggesting that adenine itself did not affect Ade4 condensates. Conversely, in *hpt1Δ* cells, in which incorporated hypoxanthine was not metabolized further, the addition of hypoxanthine failed to disassemble Ade4 foci (Fig 5B and 5D), indicating that hypoxanthine itself also did not affect Ade4 condensates. Notably, adenine failed to completely disassemble Ade4-GFP foci in *hpt1Δ* cells, suggesting that adenine affected wild-type cells primarily through its metabolic conversion to hypoxanthine.

We also investigated the effects of purine nucleotide metabolism on condensate disassembly. In *ade12Δ* cells, in which IMP was not converted to AMP via adenylosuccinate (S-AMP), the addition of hypoxanthine failed to disassemble Ade4 foci, whereas adenine moderately decreased foci (Fig 5B and 5E). The inhibition of Ade12 by alanosine, a specific inhibitor of adenylosuccinate synthetase [33], produced similar results (S9A–S9C Fig). Therefore, we conclude that the anabolism of external purine bases is required to regulate the assembly of Ade4 condensates.

### Purine nucleotides attenuate the assembly of Ade4 condensates in vitro

The results of our genetic analyses suggest that AMP and related purine nucleotide derivatives play a vital role as functional metabolites that regulate the formation of Ade4 condensates. To directly test this possibility, we examined the inhibitory effects of purine nucleotides on the in vitro assembly of Ade4-mNG condensates. In the presence of GMP, AMP, ADP, and ATP, the condensation of Ade4-mNG was significantly reduced (Figs 6A, 6B, and S10C). AMP, ADP, and ATP were more effective than GMP, while S-AMP did not exert a significant effect. Concentrations higher than 250 μM of AMP, ADP, or ATP

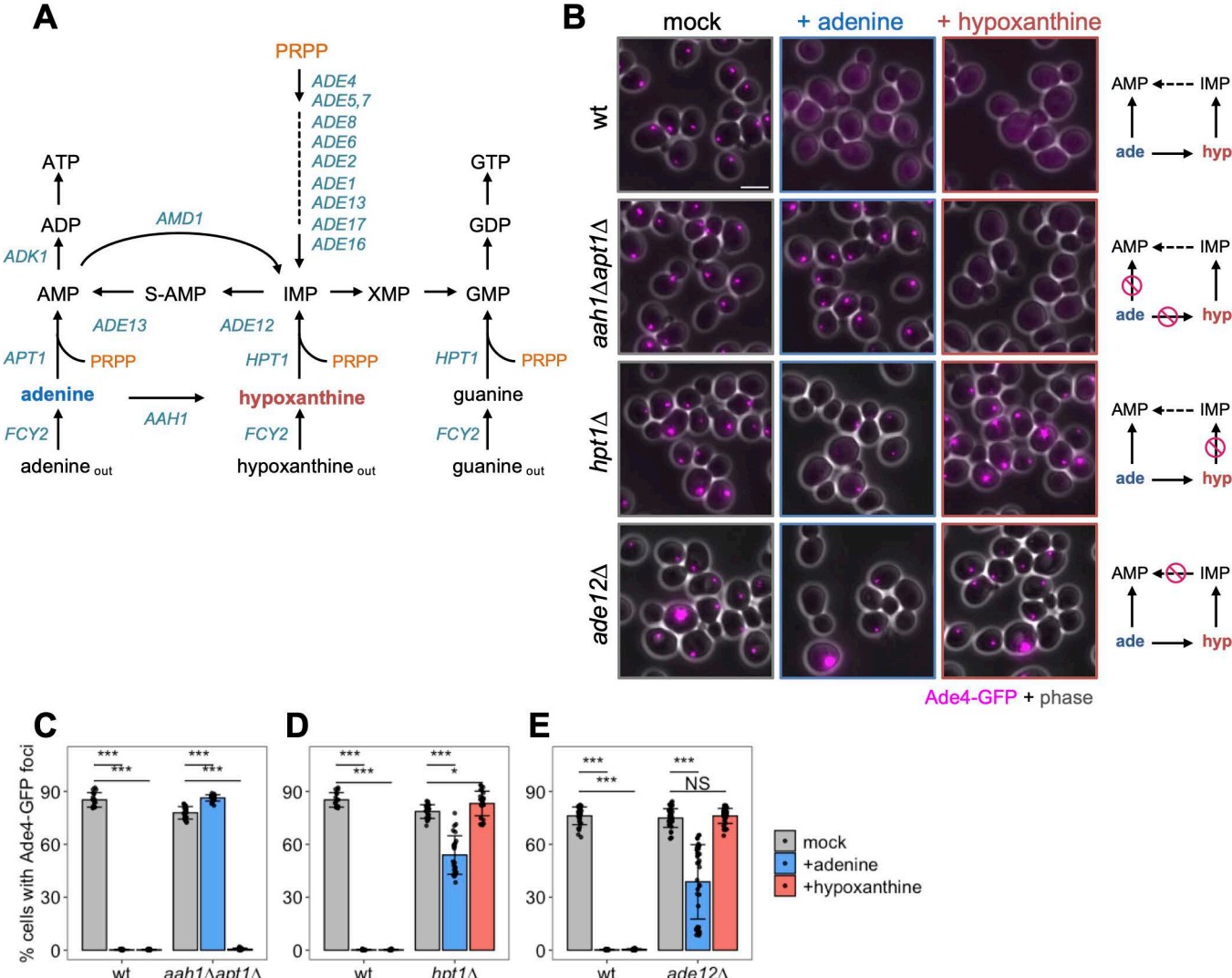

**Fig 5. Intracellular derivatives of purine nucleotides regulate the formation of Ade4 condensates.** (A) De novo and salvage pathways for purine synthesis in budding yeast. (B) Assembly of Ade4-GFP foci in purine metabolism mutants. Cells with Ade4-GFP foci in the absence of purine bases (mock) were treated with the same medium or medium containing 20-μg/mL adenine (+adenine) or 25-μg/mL hypoxanthine (+hypoxanthine) for 20 min and then imaged. Simplified views of the metabolism of the incorporated purine bases in the strains are shown. (C–E) Quantification of data shown in (B). The percentage of cells with foci per field of view was quantified and plotted. Bars and error bars indicate the mean ± 1 SD. Data were pooled from 2 to 3 independent experiments, and more than 16 fields of view were imaged for each condition. *P*-values vs. the mock were calculated using the two-tailed Steel's multiple comparison test: *$p < 0.01$; ***$p < 10^{-10}$. Scale bar = 5 μm. The data for panels **C–E** may be found in S1 Data.

were required to achieve maximal inhibitory effects, whereas the inhibitory effects of GMP plateaued at 100 μM (Figs 6C and S10D–S10F). Consistent with our in vivo results (Fig 5), adenine did not affect the in vitro condensation of Ade4 (S10A and S10B Fig). Similarly, the ribose-bound hypoxanthine derivatives, inosine and deoxyinosine (dI), had no impact on Ade4 condensation (S10A and S10B Fig). S-adenosylhomocysteine (SAH), a derivative of S-adenosylmethionine, exerted approximately one-third the inhibitory effect of AMP (S10G and S10H Fig). Importantly, AMP also suppressed the condensation of Ade4ΔIDR-mNG (Fig 6D). These results suggest that purine nucleotides, including AMP, ADP, ATP, and GMP, suppress the condensation of Ade4 molecules by inhibiting IDR-independent self-assembly.

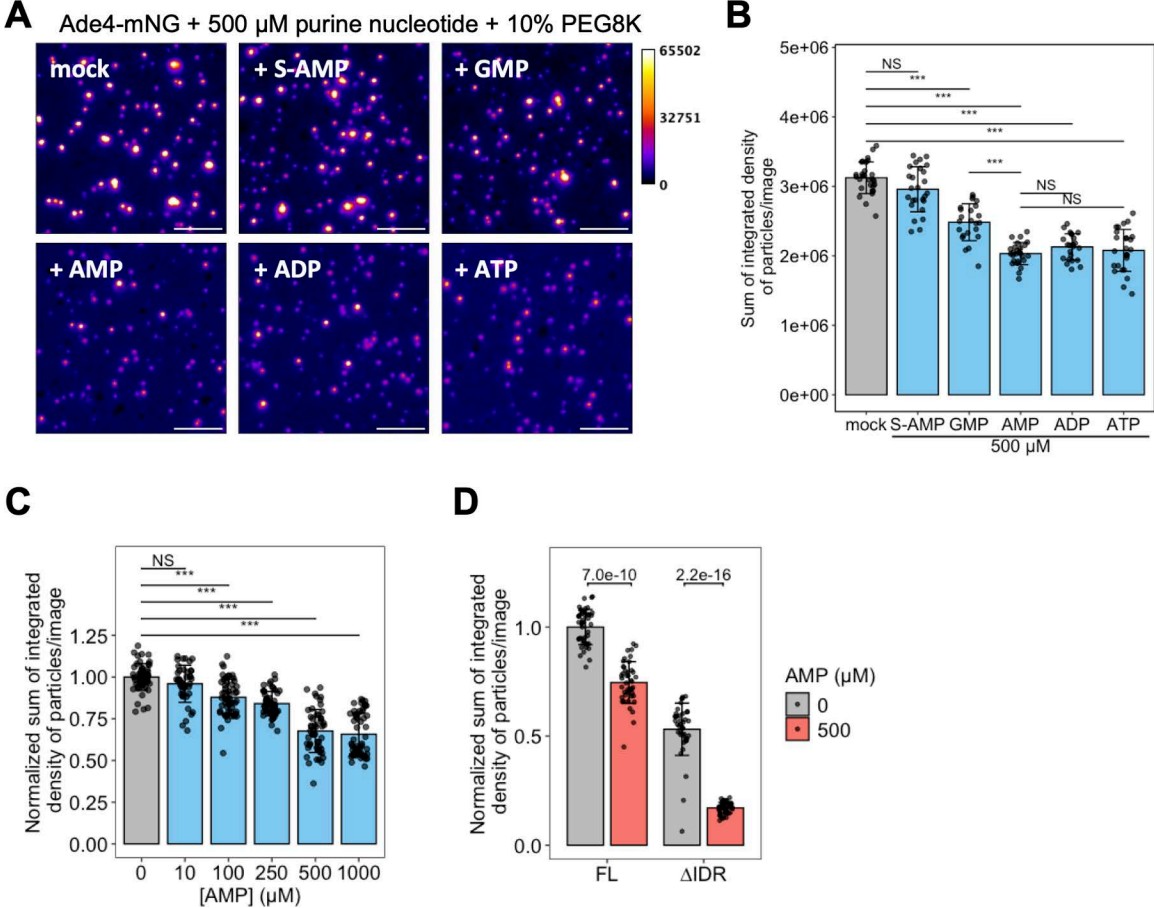

**Fig 6. Purine nucleotides attenuate the assembly of Ade4 condensates in vitro.** (A) Effects of purine nucleotides on the in vitro assembly of Ade4 condensates. Ade4-mNG (0.2 μM) was supplemented with 10% PEG8K in the presence of the indicated purine nucleotide. Fluorescence intensity is represented by pseudo-colors. Scale bars = 5 μm. (B) Quantification of data shown in (A). The integrated fluorescence intensities of condensates were summed per field and plotted. Bars and error bars indicate the mean ± 1 SD. Between 23 and 28 fields of view were imaged for each condition from two independent preparations, and an average of 1,323 particles per field were examined. *P*-values were calculated using the two-tailed Tukey-Kramer's multiple comparison test. (C) The supplementation of 0.2-μM Ade4-mNG with 10% PEG8K was performed in the presence of the indicated concentration of AMP. The integrated fluorescence intensities of the condensates were summed per field, normalized to the mean at 0 μM, and plotted. Bars and error bars indicate the mean ± 1 SD. From 4 independent preparations, 41–55 fields of view were imaged for each condition, and an average of 1,200 particles per field were examined. *P*-values were calculated using the two-tailed Steel's multiple comparison test. (D) Effects of AMP on the in vitro assembly of Ade4 and Ade4ΔIDR condensates. The supplementation of 0.2-μM Ade4-mNG (FL) or Ade4ΔIDR-mNG (ΔIDR) with 10% PEG8K was performed in the presence of 0- or 500-μM AMP. The integrated fluorescence intensities of condensates were summed per field, normalized to the mean of FL at 0-μM AMP, and plotted. Bars and error bars indicate the mean ± 1 SD. Between 40 and 45 fields of view were imaged for each condition from 3 independent preparations. *P*-values were calculated using the two-tailed Welch's *t* test. The data for panels **B–D** may be found in S1 Data.

## PRPP facilitates the assembly of Ade4 condensates in vitro and in vivo

The inhibitory effects of purine nucleotides on the in vitro formation of Ade4 condensates led us to hypothesize that these nucleotides may similarly regulate the formation of intracellular Ade4 condensates. However, the limited extent of these inhibitory effects suggested the existence of additional regulatory factors that promote the condensation of Ade4. The intracellular concentrations of these factors may decrease with adenine supplementation and increase when adenine is absent. Our correlation analysis identified several metabolites whose changes inversely correlated with intracellular adenine levels (Fig 4E). Among them, we focused on IMP, which is directly synthesized from hypoxanthine, and PRPP, which

directly interacts with Ade4 as a substrate (Fig 4D). We found that IMP did not affect the assembly of Ade4 condensates in vitro, even at concentrations that were higher than those in yeast cells [8] (S11A and S11B Fig). In contrast, PRPP concentrations higher than 50 µM significantly increased the mean fluorescence intensity and total mass of in vitro Ade4 condensates under 10% PEG conditions (Figs 7A and S12A). Quantitative analyses revealed that adenine supplementation reduced intracellular PRPP levels from ~30 µM to less than 10 µM (S8B Fig), guiding our selection of PRPP concentrations within the range of 10–200 µM for subsequent assays. Notably, the promoting effects of PRPP were more evident at lower PEG concentrations, at which Ade4 condensates rarely formed; the addition of 100-µM PRPP enabled the formation of Ade4 condensates in 5% and 7.5% PEG (Figs 7B, 7C, S12B, and S12C). These results suggest that PRPP not only enhances Ade4 condensation under condensing conditions (≥10% PEG8K) but also promotes the formation of Ade4 condensates under non-condensing conditions (≤7.5% PEG8K). Furthermore, the inhibitory effect of 1-mM AMP on Ade4 condensation was effectively counteracted by PRPP at concentrations higher than 50 µM in both 7.5% PEG8K and 10% PEG8K (Figs 7D, 7E, and S12D), suggesting that PRPP antagonizes the inhibitory effect of AMP. In contrast, in 5% PEG8K, even 200-µM PRPP failed to counteract the effect of AMP (S12E–S12G Fig).

We then examined the effects of PRPP on the formation of cellular Ade4 condensates. The 13-residue PRPP binding motif at the catalytic site is well conserved among PPATs of various species, including budding yeast, with two adjacent aspartic acids that are crucial for PRPP binding [34] (S13A Fig). A structural analysis of *Escherichia coli* PPAT showed that these aspartic acids formed hydrogen bonds with carbocyclic PRPP, a stable analog of PRPP [35] (Fig 7F). This motif also interacted with purine nucleotides in bacterial PPATs [36–38] (Figs 7G and S13B), partially explaining the competitive inhibition of the enzyme by purine nucleotides. We introduced mutations at these aspartic acids (D373A and D374A, referred to as Ade4-DA) and investigated whether the interaction with PRPP affected the assembly of Ade4 condensates. Ade4-DA-mNG was virtually incapable of forming punctate structures (Fig 7H), demonstrating that the interaction with PRPP was essential for the assembly of Ade4 condensates in cells.

Further purification of the mutant Ade4 protein allowed us to examine its in vitro biochemical properties (S13C Fig). Ade4-DA-mNG condensated into particles under 10% PEG conditions, similar to wild-type Ade4-mNG (S13D Fig), indicating that the mutation did not impair the self-assembly of Ade4 under crowding conditions. We found that PRPP only negligibly promoted the assembly of Ade4-DA-mNG condensates (Fig 7I), suggesting that the binding of PRPP at the catalytic site was responsible for facilitating the condensation of Ade4 molecules. In contrast, the DA mutant remained sensitive to AMP; AMP inhibited the assembly of Ade4-DA-mNG condensates, albeit less effectively than that of Ade4-mNG condensates (Fig 7J). Consistently, PRPP was unable to counteract the inhibitory effects of AMP on the condensation of Ade4-DA, even at concentrations higher than 500 µM (Figs 7K, S13E, and S13F). Therefore, while the interaction of purine nucleotides at the PRPP binding motif partially contributed to the inhibition of Ade4 condensation, purine nucleotides also appeared to suppress Ade4 condensation through additional binding sites (Fig 7G).

## MD calculations predict the possible condensation-dependent activation of PPAT by enabling intermolecular substrate channeling

The induction of Ade4 condensate formation by the substrate PRPP prompted us to propose that the condensation of Ade4 promotes its enzymatic activity. In glutamine-dependent PRA synthesis (Gln-PRA activity, Fig 8A), $NH_3$ is initially released from glutamine at the GATase site and must then be transferred to PRPP at the PRTase site. A structural analysis of *E. coli* PPAT suggested that PRPP binding creates a transient hydrophobic channel linking the GATase and PRTase sites [35], thereby facilitating the efficient intramolecular transfer of $NH_3$ between these catalytic sites (Figs 8B, 8C, and S14A). However, this intramolecular substrate channeling may be independent of the condensation state of the enzyme. In addition to glutamine-derived $NH_3$, PPAT also directly utilized ambient $NH_3$ as a nitrogen donor to produce PRA, referred to as $NH_3$-dependent PRA synthesis ($NH_3$-PRA activity, Figs 8A and S14B) [6]. Based on a

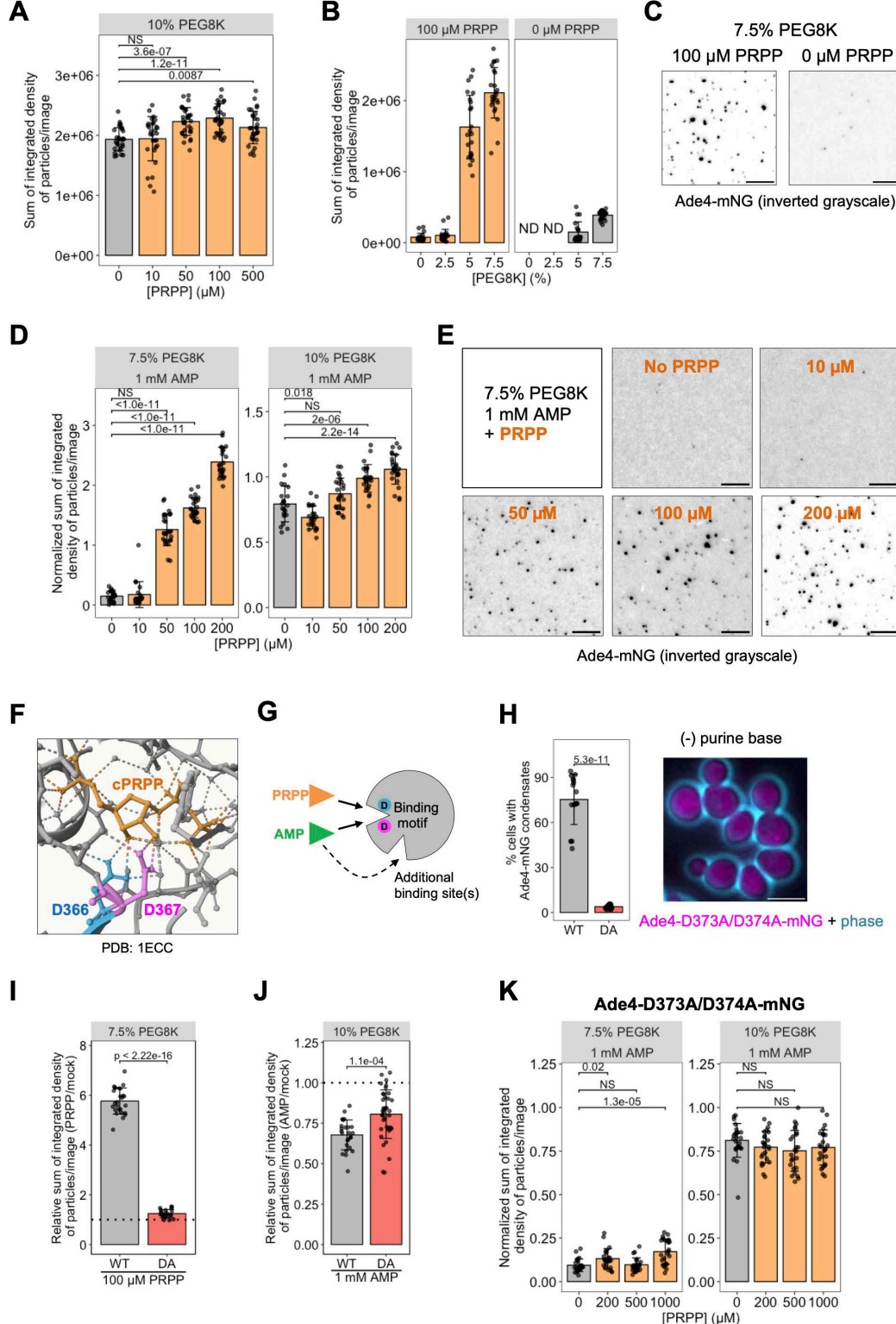

**Fig 7. PRPP facilitates the assembly of Ade4 condensates.** (A) PRPP augments the in vitro assembly of Ade4 condensates. The supplementation of 0.2-µM Ade4-mNG with 10% PEG8K was performed in the presence of the indicated concentration of PRPP. The integrated fluorescence intensities of the condensates were summed per field and plotted. Throughout the figure, bars and error bars indicate the mean ± 1 SD. From 2 independent

preparations, 29–31 fields of view were imaged for each condition, and an average of 1,200 particles per field were examined. *P*-values were calculated using the two-tailed Steel's multiple comparison test. **(B,** C) PRPP promotes the formation of Ade4 condensates in vitro at suboptimal PEG concentrations. In the presence of 100- or 0-μM PRPP, 0.2-μM Ade4-mNG was supplemented with PEG8K at the indicated concentration and imaged. The integrated fluorescence intensities of the condensates were summed per field and plotted in (B). From 2 independent preparations, 19–24 fields of view were imaged for each condition. ND, not determined. Representative images at 7.5% PEG are shown in (C). **(D,** E) PRPP antagonizes the inhibitory effects of AMP. In the presence of 0- or 1-mM AMP and the indicated concentrations of PRPP, 0.2-μM Ade4-mNG was supplemented with 7.5% or 10% PEG8K. The integrated fluorescence intensities of the condensates were summed per field, normalized to the mean of the mock control (no AMP or PRPP), and plotted in (D). From 2 independent preparations, 21–27 fields of view were imaged for each condition. *P*-values were calculated using the two-tailed Steel's multiple comparison test. Representative images at 7.5% PEG are shown in (E). (F) Interactions between a PRPP analog and specific amino acid residues in the PRPP binding motif in the *E. coli* PPAT (pdb file: 1ecc). The dotted line indicates a hydrogen bond. (G) PRPP and purine nucleotides (*e.g.*, AMP) interact competitively with the binding motif. Purine nucleotides may also interact with additional binding site(s) and inhibit Ade4 condensation. See the text for details. (H) The interaction of Ade4 with PRPP is important for condensate assembly. Ade4-mNG (WT) and Ade4-D373A/ D374A-mNG (DA) cells were incubated in the absence of purine bases for 45 min and imaged. The percentages of cells with condensates per one field of view were quantified and plotted. Data were pooled from 2 independent experiments, and more than 16 fields of view were imaged for each condition. *P*-values were calculated using the two-tailed Mann–Whitney U test. A representative image of mutant cells is shown. (I) PRPP did not stimulate the condensate assembly of the DA mutant. The supplementation of 0.2-μM Ade4-mNG (WT) and Ade4-D373A/D374A-mNG (DA) with 7.5% PEG8K was performed in the presence of 0- or 100-μM PRPP and imaged. The integrated fluorescence intensities of the condensates were summed per field, normalized to the mean at 0-μM PRPP (dotted line), and plotted. From 2 independent preparations, >22 fields of view were imaged for each condition. *P*-values were calculated using the two-tailed Welch's *t* test. (J) The DA mutant is less sensitive to the inhibitory effects of AMP. The supplementation of 0.2-μM Ade4-mNG (WT) and Ade4-D373A/D374A-mNG (DA) with 10% PEG8K was performed in the presence of 0- or 1-mM AMP and imaged. The integrated fluorescence intensities of the condensates were summed per field, normalized to the mean at 0-mM AMP (dotted line), and plotted. From 2 to 3 independent preparations, >24 fields of view were imaged for each condition. *P*-values were calculated using the two-tailed Mann–Whitney U test. (K) PRPP did not antagonize AMP in the condensate formation of the DA mutant. In the presence of 0- or 1-mM AMP and the indicated concentrations of PRPP, 0.2-μM Ade4-D373A/D374A-mNG was supplemented with 7.5% or 10% PEG8K. The integrated fluorescence intensities of the condensates were summed per field, normalized to the mean of the mock control (no AMP and no PRPP), and plotted. From 2 independent preparations, 22–25 fields of view were imaged for each condition. *P*-values were calculated using the two-tailed Steel's multiple comparison test. Scale bars = 5 μm. The data for panels **A, B, D, E,** and **H–K** may be found in S1 Data.

condensation-dependent activation mechanism, we hypothesized that the Ade4 condensate may enable the intermolecular transfer of $NH_3$ between molecules (Fig 8C, right).

To test this hypothesis, we investigated the behavior of $NH_3$ on the PPAT molecule by conducting MD simulations using the crystal structure of *E. coli* PPAT in its active form [35]. MD simulations for the holo state of PPAT, which included glutamate, $NH_3$, PRPP, and $Mg^{2+}$, showed that $NH_3$ mostly remained at the initial or adjacent positions during 10 trials and did not reach the catalytic site, despite the presence of the hydrophobic channel (Fig 8D). This result suggests a low frequency of intramolecular channeling relative to our simulation time. In contrast, in the semi-holo state containing only glutamate and $NH_3$, $NH_3$ escaped through the hydrophobic channel once in 10 trials. Unexpectedly, $NH_3$ was instead released from the molecule through a transient channel directed toward Arg333 in 4 out of 10 trials (Fig 8E and 8F and S8 Movie). This Arg333 is conserved in yeast Ade4 and is referred to as the front channel. Prior to the release of $NH_3$, we noted that the hydrophobic side chain of Phe260 flipped from its initial structure. Furthermore, Arg333 functioned as a gate for $NH_3$ by forming hydrogen bonds with the backbone oxygens of Tyr259, Phe260, and His26 and the phosphate group of PRPP (Fig 8G). Notably, the absence of PRPP increased structural fluctuations in the Arg gate in the semi-holo state, thereby increasing the likelihood of gate opening.

Additional simulations on the Arg333Ala mutant under holo conditions confirmed the critical role of this Arg gate. The flipping of Phe260 and subsequent release of $NH_3$ were observed in 6 out of 10 trials on the Arg333Ala mutant, despite the presence of PRPP at the catalytic site (Fig 8H). This increased gate-opening frequency in the mutant suggests that the normally closed Arg gate, coupled with PRPP, suppresses the flipping of the Phe260 side chain and the release of $NH_3$. Since the PRTase catalytic site utilizes ambient $NH_3$ as a nitrogen donor [6], the release of PRA may simultaneously facilitate the release of internal glutamine-derived $NH_3$ through the front channel (Figs 8I and S14C). Collectively, these simulation results support PPAT condensation facilitating the intermolecular transfer of leaked $NH_3$ between closely located molecules, leading to the efficient utilization of glutamine-derived $NH_3$ (Fig 8C, right panel and S14D Fig).

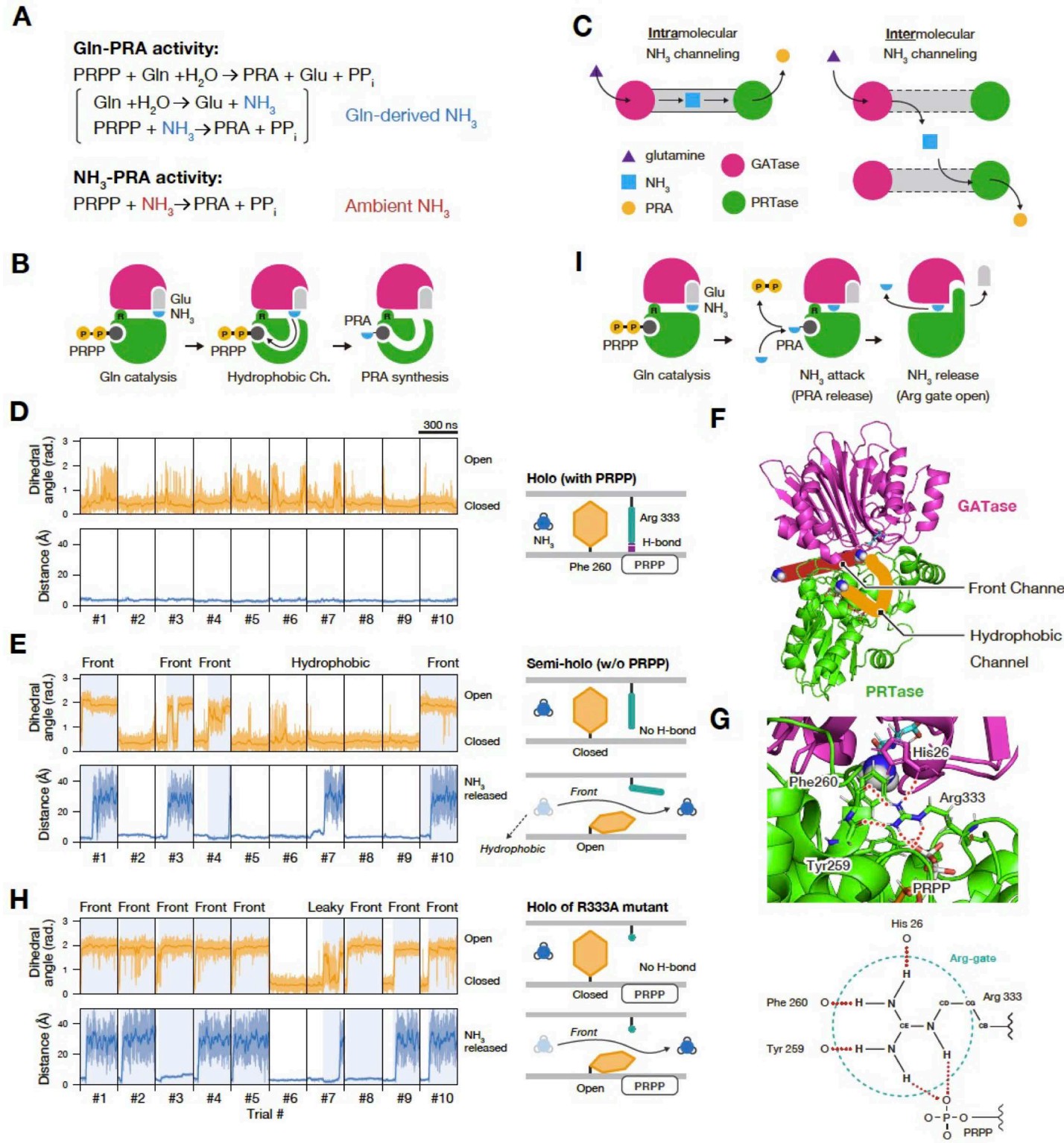

**Fig 8. MD calculations predict the possible condensation-dependent activation of PPAT by enabling intermolecular substrate channeling.** (A) Reaction formulae corresponding to the glutamine-dependent PRA synthesis (Gln-PRA activity) and $NH_3$-dependent PRA synthesis ($NH_3$-PRA activity) of PPAT. (B) Reaction mechanisms suggested in a previous study. (C) Intramolecular and intermolecular channeling of glutamine-derived $NH_3$. **(D, E,**

H) Time series for the dihedral angle of the Phe260 side chain and the distance of $NH_3$ from the initial structure (**D**: holo; **E**: semi-holo; and H: R333A mutant). Moving averages are plotted in bold lines. Under each condition, 300-ns simulations were performed 10 times individually. Blue backgrounds indicate the open state of the Phe260 sidechain. The schematic diagram shows the state of the Arg gate and the movement of the $NH_3$ molecule. (F) The hydrophobic (orange) and front (red) channels of *E. coli* PPAT. (G) The conserved Arg residue functions as a gate of the front channel. Top: A structural snapshot around the Arg gate. Bottom: A scheme of the Arg gate. Hydrogen bonds are shown as red dotted lines. (I) The reaction mechanisms suggested in the present study. The data for panels **D, E**, and **H** may be found in S1 Data.

### Formation of Ade4 condensates is important for efficient DPS during cell growth

Authentic Gln-PRA activity and $NH_3$-PRA activity (Fig 8A) have both been implicated in the de novo biosynthesis of purine nucleotides in vivo [39,40]. Since $NH_3$-PRA activity depends on the concentration of $NH_4^+$ in medium, a fraction of which is present as $NH_3$ [6], Ade4 condensates may be critical for Gln-PRA activity under low ammonium conditions. To examine this possibility, we compared the growth of cells expressing wild-type Ade4 with those expressing Ade4ΔIDR, which is unable to form condensates in vivo. On solid medium without adenine, the growth of 3×HA-tagged Ade4ΔIDR cells was markedly slower than that of wild-type Ade4 cells (Fig 9A). The protein level of Ade4ΔIDR grown in adenine-free medium was similar to that of wild-type Ade4 (Fig 9B). Supplementation with adenine restored the growth of the *ADE4ΔIDR* mutant to wild-type rates, indicating that the slow growth of the mutant was due to reduced enzyme activity rather than dominant-negative inhibition by the mutant protein. Notably, the growth of the *ADE4ΔIDR* mutant significantly decreased with reductions in the concentration of ammonium ions from ammonium sulfate (AS) in the medium (Fig 9A). We further quantified cell growth by measuring growth curves in liquid medium. The *ADE4ΔIDR* mutant exhibited significantly slower growth, particularly in the absence of ammonium ions, while the growth rate of the wild-type strain remained almost independent of the ammonium ion concentration in the medium (Fig 9C and 9D). Collectively, these results suggest that the condensation of Ade4 into particles is critical for DPS, particularly under low ammonium conditions, where Gln-PRA activity plays a pivotal role.

To further confirm whether the aggregation state of the Ade4 protein affects DPS, we compared the growth of cells expressing Ade4ΔIDR-GFP and Ade4ΔIDR-mGFP (S15A–S15C Fig). As controls, we also measured the growth of cells expressing Ade4-GFP and Ade4-mGFP. Growth rates in the absence of adenine, ranked in descending order, were as follows: Ade4-GFP ≥ Ade4-mGFP> Ade4ΔIDR-GFP> Ade4ΔIDR-mGFP (S15C and S15D Fig). The difference between Ade4-GFP and Ade4-mGFP was small, whereas that between Ade4ΔIDR-GFP and Ade4ΔIDR-mGFP was significant (S15D Fig). The expression levels of the four constructs were similar (S15E Fig). As shown in Figs 1B and 1C, GFP dimerization artificially promoted Ade4 aggregation. Therefore, these results indicate that the slower growth of the *ADE4ΔIDR* mutant was partially restored by the aggregation-promoting property of GFP, again supporting the hypothesis that Ade4 condensation enhances DPS in cells.

### Discussion

We herein demonstrated that the budding yeast PPAT forms intracellular condensates through TORC1-mediated macromolecular crowding and PRPP-dependent self-assembly to regulate DPS in response to extracellular purine bases (Fig 10). MD simulations revealed a previously unidentified pathway by which an intermediate $NH_3$ derived from glutamine leaks from the PPAT molecule (Figs 8I and S14C). We propose that Ade4 condensation enhances Gln-PRA activity by facilitating a domino-like chain reaction through intermolecular intermediate channeling. The condensation of PPAT, which has two catalytic domains, is considered to be equivalent to the co-clustering of two distinct enzymes, GATase and PRTase, in a confined space. Although $NH_3$ is expected to diffuse rapidly throughout the cytoplasm [41], within Ade4 condensates, where the enzyme concentration is high, another enzyme may capture $NH_3$ and effectively process it by the PRTase domain (Fig 8C, right panel and S14D Fig). This form of enzyme activation by intermolecular $NH_3$ channeling

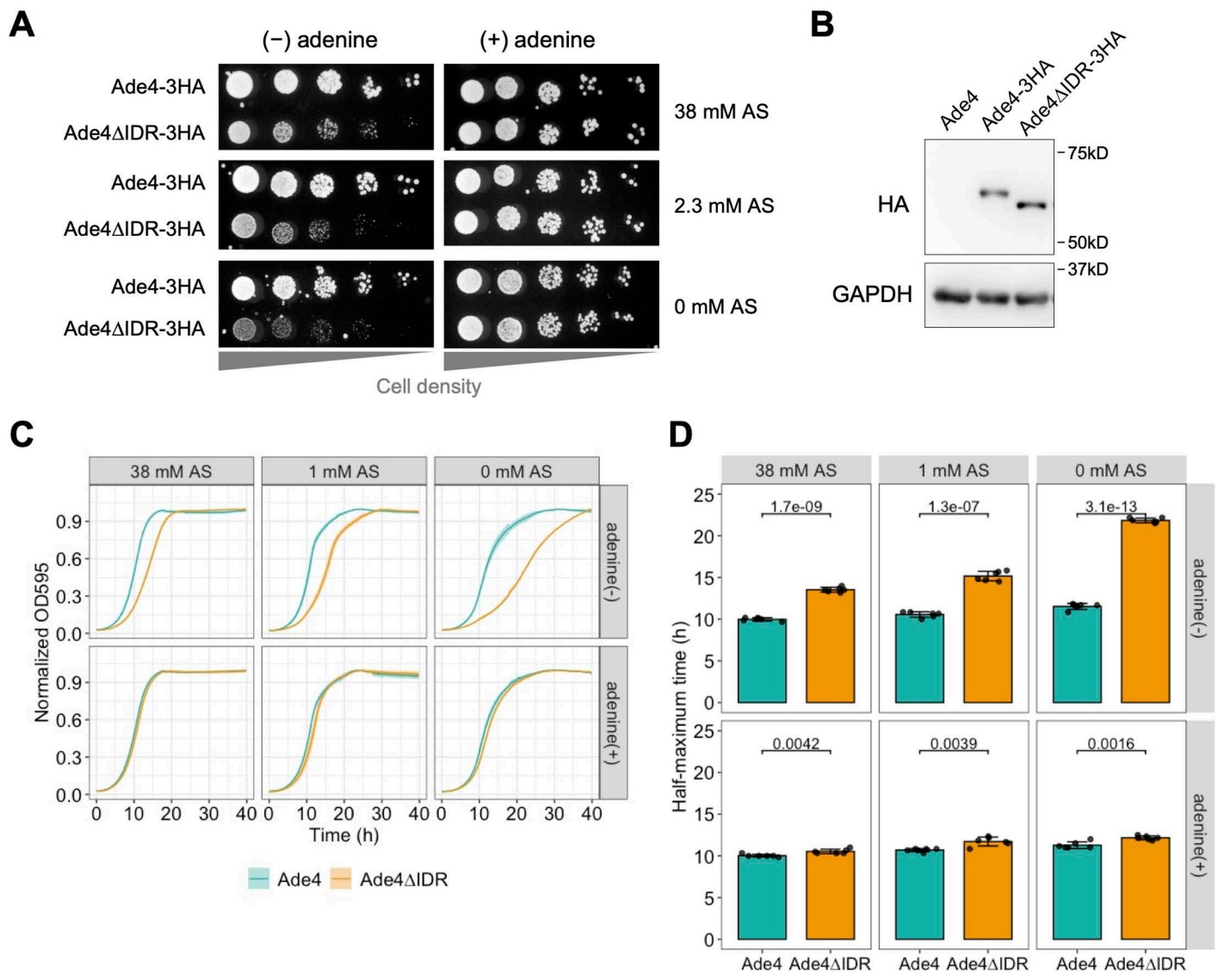

**Fig 9. The formation of Ade4 condensates is important for efficient DPS during cell growth.** (A) Growth of cells expressing Ade4−3HA and Ade4ΔIDR-3HA on solid media. Cells of each strain were spotted on medium containing the indicated concentration of AS, and then grown at 30°C for 2–3 days. (B) Immunoblot analysis of the protein level of Ade4. Total cell extracts were prepared from cells expressing the indicated Ade4 construct. Ade4−3HA and Ade4ΔIDR-3HA were detected as ~60-kDa bands by an anti-HA antibody. The bands of GAPDH are shown as a loading control. (C) Growth curves of cells expressing Ade4−3HA and Ade4ΔIDR-3HA in liquid media without adenine. Cells were grown at 30°C for 40 h in medium containing the indicated concentration of AS in the absence or presence of adenine. Data are the mean of 6 replicates, and the shaded area indicates ±1 SD. (D) Quantification of data shown in (C). The time to half-maximum density (corresponding to normalized $OD_{595}$ = 0.5) was calculated and plotted. Bars and error bars indicate the mean ± 1 SD. P-values were calculated using the two-tailed Welch's t test. Uncropped images of immunoblots are available in S1 Raw images. The data for panels **C–D** may be found in S1 Data.

corresponds to clustering-mediated metabolic channeling theoretically predicted by Castellana and colleagues [42]. The present study is the first demonstration that a native enzyme may be activated by this type of metabolic channeling in eukaryotic cells.

Recent studies revealed that many metabolic enzymes form intracellular condensates by phase separation, allowing cellular metabolism to adapt to variable growth conditions [9]. These condensates serve as temporary stores for inactive

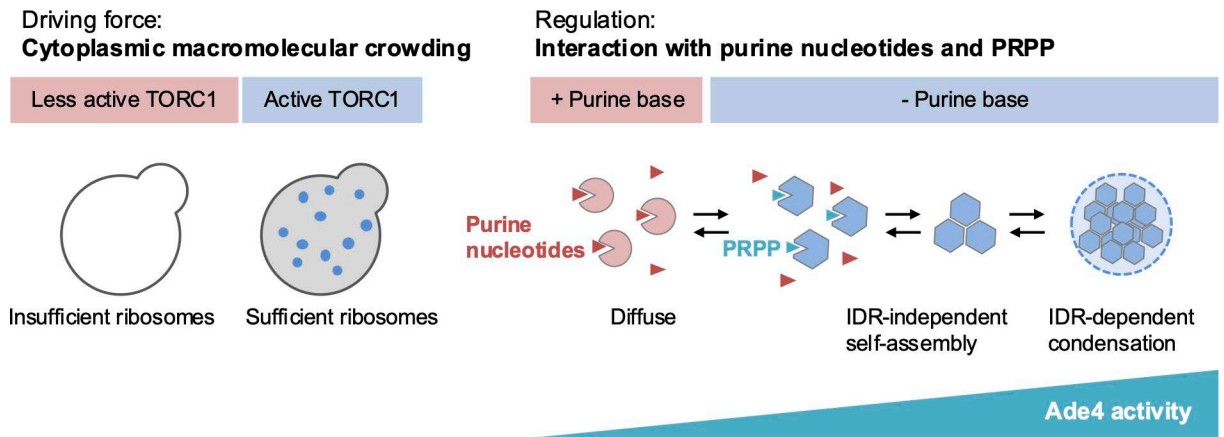

**Fig 10. Molecular crowding and interactions with effectors regulate Ade4 condensation.** A schematic model showing the regulation of Ade4 condensation proposed in the present study. See the text for details.

enzymes, helping cells survive adverse environmental conditions [9]. In some cases, filament formation by the enzyme enhances catalytic activity [43,44]. Our results add another example where the condensation of a metabolic enzyme boosts catalytic activation through intermolecular substrate channeling. Notably, Ade4 condensates are unique among the cellular condensates reported to date due to their dynamic nature; they assemble and disassemble quickly and reversibly in response to environmental purine bases and exhibit high motility.

Ade4 (PPAT) condensates resemble enzyme clusters known as purinosomes, which form reversibly in HeLa cells under purine deprivation [13]. Purinosomes enhance the channeling of metabolic intermediates in the de novo pathway near mitochondria, thereby increasing pathway flux [14]. However, these two structures differ in several respects. Purinosomes contain all six enzymes involved in DPS with formylglycinamide ribonucleotide synthase as the core, whereas cellular clusters of PPAT are smaller and only some colocalize with purinosomes [45]. Furthermore, purinosomes are static, assembling and disassembling in ~1.5 and 1 h, respectively, and are non-motile [13]. Therefore, we hypothesize that PPAT condensates are structurally distinct from purinosomes and may coexist inside or outside of them. Since PPAT is the rate-limiting enzyme in DPS, its condensates may provide a rapid and flexible response to fluctuations in environmental purine base levels, whereas purinosomes may represent a more constant adaptation to purine scarcity.

We also found that purified Ade4 self-assembled into fine particles under molecular crowding conditions. In vitro phase separation assays further demonstrated that Ade4 condensation involved both IDR-independent self-assembly and IDR-dependent condensation. In addition to the C-terminal IDR, the Ade4 protein has a short putative IDR at the N terminus. Plant and insect PPATs have an IDR at the N terminus, but not at the C terminus, suggesting that both ends of the protein contribute to condensate formation by phase separation. Therefore, multiple IDRs may be involved in complete condensate formation. Nevertheless, it is intriguing to speculate that C-terminal IDR-independent self-assembly is essential for subsequent IDR-dependent condensation, where phase separation is required for stable condensation into particles (see Fig 10, right panel).

The enzymatic activity of PPAT is considered to be regulated in a feedback manner by millimolar levels of AMP and GMP [46]. However, intracellular purine mononucleotide concentrations are generally maintained in the micromolar range, from tens to hundreds, and are not markedly affected by extracellular purine bases (Fig 4D and previous studies [47–49]). Therefore, this feedback regulation appears to be physiologically irrelevant, at least in response to environmental purine bases. In contrast, the present results indicate that purine nucleotides inhibit the IDR-independent self-assembly of Ade4, while PRPP promotes condensate formation in vitro. We speculate that these two regulators control the assembly of

Ade4 condensates, thereby affecting PPAT enzyme activity in vivo. Since ADP and ATP, similar to AMP, suppressed the assembly of Ade4 condensates, GDP and GTP may also exert similar inhibitory effects to those of GMP. Therefore, the concentrations of all purine nucleotides may be physiologically relevant and sufficient to exert inhibitory effects on Ade4 condensate formation in cells.

The intracellular concentrations of purine nucleotides were previously shown to slightly increase in the presence of environmental purine bases [47–49], while that of PRPP decreased to ~10 μM. At this level, PRPP is unable to interact with Ade4 in order to promote its assembly, and, thus, purine nucleotides may predominantly interact with Ade4 and prevent condensate formation. In the absence of purine bases, PRPP levels may increase by up to several tens of micromolar, thereby promoting the condensation of Ade4. Since PRPP-docked Ade4 is prone to condensation, the local concentration of PRPP within condensates is expected to markedly increase. Consistent with this notion, the $K_m$ of PPAT for PRPP in yeast was previously reported to be 110 μM [50], which is 3.7-fold higher than the intracellular concentration of PRPP even in the absence of adenine. Similarly, in mammals, intracellular PRPP concentrations range between 5 and 30 μM [51], while the enzyme's $K_m$ for PRPP is 480 μM [52]. The enrichment of PRPP within Ade4 condensates provides an example of the spatial control of substrate concentrations by biomolecular condensates [53].

Consistent with the present results, the binding of PRPP to the specific binding motif at the catalytic site of *E. coli* PPAT has been reported to induce a structural change in the enzyme [35,54], which may contribute to the enhanced clustering of the enzyme under molecular crowding conditions. AMP also interacts with the PRPP binding motif; however, in contrast to PRPP, it inhibits self-assembly. Therefore, competitive binding to the motif explains the mechanism by which PRPP counteracts the effects of purine nucleotides. However, since the Ade4-DA mutant remains sensitive to inhibition by AMP, further structural analyses are required to elucidate the assembly mechanisms and effects of purine nucleotides.

It is important to note that the addition of purine bases to culture media was previously shown to reduce the concentration of PRPP in both bacterial cells [51] and mammalian cells [55], suggesting a conserved mechanism across species. The most likely explanation for this is the consumption of PRPP, coupled with the conversion of incorporated purine bases by hypoxanthine-guanine phosphoribosyltransferase (HGPRT, known as Hpt1 in budding yeast). HGPRT utilizes PRPP as a substrate more efficiently than PPAT, as evidenced by its markedly lower $K_m$ for PRPP [56]. Consistently, HGPRT-negative lymphoblasts from Lesch–Nyhan syndrome patients showed 4- to 8-fold higher intracellular PRPP levels than those from normal individuals [57], which supports this notion and is consistent with the phenotype of *hpt1Δ* cells observed in the present study.

We found that ribosome biogenesis downstream of TORC1 signaling was required for Ade4 condensate formation and its maintenance. These results align with the theory that TORC1 regulates the macromolecular crowding of the cytoplasm through ribosome synthesis, which, in turn, controls biomolecular condensation by phase separation [20]. The present study is the first experimental demonstration of a metabolic enzyme assembling into condensates via the TORC1-regulated phase transition of the cytoplasm (see Fig 10, left panel). Since TORC1 and PPAT IDRs are both conserved across many species, the condensation of PPAT by TORC1 may represent a common feature in eukaryotic cells.

A recent study suggests that polysomes rather than individual ribosomes play a major role in cytoplasmic crowding in yeast cells [58]. According to this study, while the inhibition of TORC1 causes rapid polysome collapse that is completed within 15 min, it also induces the slower autophagy-dependent degradation of ribosomes that takes more than 1 h. Therefore, the rapid disassembly of Ade4 condensates by rapamycin (S4 Fig) appears to support the notion that polysomes primarily contribute to the molecular crowding of the cytoplasm. However, active ribosome synthesis is still important for cytoplasmic molecular crowding because the amounts of polysomes correlate with the number of ribosomes. Previous studies showed that a diazaborine treatment not only significantly reduced 60S ribosomal subunits but also simultaneously decreased polysomes (sometimes within 30 min) [26,59], indicating the dependence of polysome abundance on ribosome concentrations.

The involvement of TORC1 is another crucial aspect of Ade4 condensates. TORC1 is activated by sensing nutrients, such as amino acids and glucose [60]. Therefore, the requirement of TORC1 activity for PPAT condensation ensures that the de novo synthesis of purines is maximized under conditions that are favorable for cell growth. In mammals, TORC1 transcriptionally up-regulates the enzymes involved in the mitochondrial tetrahydrofolate cycle, thereby increasing DPS [61]. Correlating the metabolic flux of purine synthesis with the nutritional status via TORC1 activity appears to be a conserved and multi-layered regulatory mechanism for cell proliferation.

The assembly of Ade4 condensates requires promotion by PRPP as discussed above, suggesting that the molecular crowding effect in yeast cells is not sufficient, but marginal for the spontaneous condensation of the Ade4 protein itself. The conditions for Ade4 condensation in vivo and in vitro are summarized in a table (S16 Fig), which shows that the 7.5% PEG8K condition most accurately reflects the regulation of Ade4 condensation in cells, suggesting that the molecular crowding effect in the cytoplasm of yeast cells with activated TORC1 activity corresponds to that PEG8K concentration. In 2.5% PEG8K, the molecular crowding effect is so weak that even PRPP is unable to induce Ade4 condensation, which may correspond to intracellular crowding in cells treated with rapamycin or diazaborine. In 5% PEG8K, the presence of PRPP may switch Ade4 condensation only in the absence of AMP, a non-physiological condition. In 10% PEG8K, the molecular crowding effect is so strong that condensation occurs irrespective of PRPP or AMP. Appropriate molecular crowding conditions that are not too strong or weak may enable the regulation of Ade4 condensation by PRPP.

Disorders in the regulation of purine metabolism are associated with various diseases, such as cancer, hyperuricemia (gout), and immunological defects [62,63]. A more detailed examination of purine metabolism will provide insights into the molecular mechanisms underlying these diseases. Importantly, a recent study implicated the up-regulation of PPAT in the malignant progression of many cancer types [64]. Therefore, elucidating the regulatory mechanism of PPAT activity will not only enhance our understanding of purine metabolism in a biological context but also contribute to advances in basic and therapeutic research on cancer.

While we hypothesize that IDR, being small and separate from enzymatic domains, does not directly impair the enzymatic activity of PPAT, we lack methods to precisely measure and compare the enzymatic activities of Ade4 and Ade4ΔIDR proteins, particularly under molecular crowding conditions. This indicates that the slow growth of the *ADE4ΔIDR* mutant is due to a reduction in intrinsic enzyme activity, the lack of activation by a condensation-dependent mechanism, or both. However, we consider the condensation-induced activation of Ade4 to play a significant role in purine synthesis in order to support cell growth because the artificial aggregation of Ade4ΔIDR by GFP dimerization partially restored the growth defect of the *ADE4ΔIDR* mutant (S15 Fig). Further biochemical analyses of the enzymatic activity of Ade4 condensates may resolve this issue.

## Materials and methods

### Yeast strains and plasmids

The yeast strains, plasmids, and primers used in the present study are listed in S4, S5, and S6 Tables, respectively. These strains were constructed by a PCR-based method [65] and genetic crosses. The yeast knockout strain collection was originally purchased from GE Healthcare (cat. # YSC1053). All strains and plasmids are available from the Yeast Genetic Resource Centre Japan (https://yeast.nig.ac.jp/yeast/top.jsf).

Regarding the construction of plasmids used for the C-terminal tagging of monomeric GFP or mKate2, DNA fragments encoding the corresponding fluorescent proteins were artificially synthesized (Eurofin Genomics, Ebersberg, Germany). These fragments were replaced with GFP of pYM25 [65] using the *Sal*I/*Bgl*II sites to yield MTP3114 and MTP3131, respectively. The plasmid used for the C-terminal tagging of mNG-Strep tag II (MTP3139) was constructed in the same manner: a DNA fragment encoding single mNG fused with the Strep II tag (WSHPQFEK) by a spacer sequence (ELYKGSA) was synthesized and then replaced with the GFP of pYM25.

## Media and cell culture

Synthetic medium optimized for the growth of the BY strain was prepared as described by Hanscho and colleagues [66]. Growth assays in solid and liquid media used optimized synthetic media lacking glutamate, phenylalanine, serine, and threonine. Yeast nitrogen base without amino acids, but with AS (cat. # Q30009) was purchased from Invitrogen. Yeast nitrogen base without amino acids and AS (cat. # CYN0501) was obtained from FORMEDIUM. Synthetic medium without ammonium sulphate was supplemented with 8-mM proline as an alternative nitrogen source. Rich medium was standard YPD medium composed of 1% (w/v) yeast extract (BD, cat. # 212750), 2% (w/v) bacto-peptone (BD, cat. # 211677), and 2% (w/v) D-glucose. Glucose (cat. # 049–31165), adenine sulfate (cat. # 018–10613), hypoxanthine (cat. # 086–03403), 1, 6-hexanediol (cat. # 087–00432), and digitonin (cat. # 043–2137) were purchased from FUJIFILM Wako. Cells were grown to the mid-log phase at 30°C before imaging unless otherwise noted. To replace medium containing adenine with medium lacking adenine, cells were collected in a microtube by brief centrifugation and suspended in medium lacking adenine. This washing procedure was repeated at least twice. Rapamycin was purchased from LC Laboratories (cat. # R-5000).

## Microscopy

Cells expressing a GFP-fusion protein were concentrated by centrifugation and sandwiched between a slide and coverslip (No. 1.5 thickness, Matsunami, Osaka, Japan). Immobilized cells were imaged using an inverted fluorescent microscope (Eclipse Ti2-E, Nikon, Tokyo, Japan) equipped with a CFI Plan Apoλ 100× Oil DIC/NA1.45 objective lens and CMOS image sensor (DS-Qi2, Nikon). The fluorescent signal was collected from stacks of 11 $z$-sections spaced by 0.5 μm, and the maximum projections of optical sections were shown unless otherwise noted. A CFI Plan Apoλ 60× Oil Ph3 DM/NA1.40 or CFI Plan Apoλ 100× Oil Ph3 DM/NA1.45 objective lens was used for phase contrast imaging. Images of cells were acquired from several fields of view for each experimental condition. All digital images were processed with the Fiji image platform [67].

## Quantitative analysis of intracellular fluorescent condensation

Intracellular fluorescent condensation (*i.e.*, Ade4-GFP foci or Ade4-mNG particles) was automatically detected and analyzed by running Fiji macros (https://github.com/masaktakaine/FP-foci-quantification and https://github.com/masaktakaine/FP-granules-quantification). Briefly, these macros extract cell outlines from phase-contrast images and detect fluorescent condensation inside the cell boundary using the *FindMaxima* function of Fiji software.

To monitor temporal changes in the number of intracellular Ade4-mNG particles by time-lapse imaging, the number of maxima of green fluorescence intensity in the field was counted in each time frame and divided by the total cell number by running a Fiji macro (https://github.com/masaktakaine/Counting-FP-granules-in-time-lapse). In cases in which phase-contrast images were unavailable, we manually examined condensation.

## Spot growth assay

Cells were grown to the mid-log phase at 30°C in synthetic medium, and cell density (cells/mL) was measured by an automated cell counter (TC20, Bio-Rad). The culture was diluted by medium lacking nitrogen and carbon sources to a density of $1 \times 10^6$ cells/mL, serially diluted 5-fold, and spotted on plates. Plates were incubated at 30°C for 2–3 days and then imaged by an image scanner (GT-X830, EPSON).

## Measurement of growth curves in liquid cultures

Cell growth in liquid media was monitored in a 96-well flat-bottomed plate using a microplate reader (iMark, Bio-Rad). Cells were grown to the mid-log phase and then diluted to $OD_{600}$ = 0.1. Each well was filled with 300 μL of the cell dilution and $OD_{595}$ was measured at 30°C every 10 min.

## Immunoblot analysis

A denatured whole-cell extract was prepared according to the method reported by von der Haar [68]. Briefly, mid-log cells were harvested from 2 to 3 mL of the culture by centrifugation. The cell pellet was resuspended in a small volume (<100 µL) of lysis buffer (0.1-M NaOH, 50-mM EDTA, 2% (w/v) SDS, and 2% (v/v) β-mercaptoethanol) and then incubated at 90°C for 10 min. The sample was neutralized by the addition of a 1/10th volume of 1 M acetic acid, incubated at 90°C for 10 min, and mixed with a one-third volume of 4×Laemmli protein sample buffer (cat. # 1610747, Bio-Rad). Following centrifugation at 20,000g at 4°C for 10 min, the upper 90% of the supernatant was collected as the clarified whole-cell extract and stored at -30°C until used. Proteins were resolved by SDS-PAGE using a pre-cast gel containing 10% (w/v) acrylamide (SuperSep Ace, cat. # 195−14951, FUJIFILM Wako) and then transferred onto a PVDF membrane (Immobilon-P, Millipore) using a protein transfer system (Trans-Blot Turbo Transfer System, Bio-Rad). HA-tagged Ade4 proteins were detected using an anti-HA antibody (mouse monoclonal (12CA5), cat. # GTX16918, Funakoshi) at a 1:1,000 dilution. GFP-tagged proteins were detected using an anti-GFP antibody (mouse monoclonal (1E10H7), cat. #66002−1-Ig, ProteinTech) at a 1:1,000 dilution. GAPDH was detected using an anti-GAPDH antibody (mouse monoclonal (1E6D9), cat. # 60004−1-Ig, ProteinTech) at a 1:1,000 dilution. PGK1 was detected using an anti-PGK1 antibody (mouse monoclonal (22C5D8), cat. # ab113687, Abcam) at a 1:1,000 dilution. An HRP-conjugated secondary antibody (from sheep, cat. #NA931V, GE Healthcare) was probed at a 1:10,000 dilution. Immunoreactive bands were luminescent in a chemiluminescent substrate (Western BLoT Quant HRP substrate, cat. # T7102A, TaKaRa) and imaged using an image analyzer (LAS4000, GE Healthcare).

To analyze the phosphorylation of Rps6, proteins were resolved by SDS-PAGE using a pre-cast gel containing 12.5% (w/v) acrylamide (SuperSep Ace, cat. # 199−14971, FUJIFILM Wako). Membranes were blocked with 3% (w/v) gelatin (cat. #16605−55, Nacalai). Total Rps6 and phosphorylated Rps6 were detected using an anti-Rps6 antibody (rabbit polyclonal, cat. #ab40820, Abcam) and phospho-S6 ribosomal protein (Ser235/236) antibody (rabbit polyclonal, cat. #2211, Cell Signaling) at a 1:1,000 dilution, respectively. HRP-conjugated anti-rabbit IgG (from goat, cat. #7074S, Cell Signaling) was probed at a 1:2,000 dilution. These antibodies were diluted with Can Get Signal Immunoreaction Enhancer Solution (cat. #NKB101, Toyobo). Total Rps6 and phosphorylated Rps6 were detected on two separate blots loaded with the same samples.

## Protein purification from yeast cells

Ade4-mNG, Ade4ΔIDR-mNG, and mNG were C-terminally tagged with a single Strep-tag II, expressed in budding yeast cells and purified as follows. Yeast cells expressing the tagged protein were grown in liquid YPD medium supplemented with 0.02-mg/mL adenine sulfate at 30°C for at least 20 h. The pre-culture was diluted by 100- to 1,000-fold in 100 mL of fresh YPD plus adenine medium, and cells were further grown at 30°C to $4 \times 10^7$ cells/mL. Cells were harvested by centrifugation at 2,000g and washed twice with 1 mL of extraction buffer (EB) (0.15-M NaCl, 1-mM EDTA, 1-mM dithiothreitol, and 40-mM Tris-HCl, pH 7.5) supplemented with 1-mM PMSF and complete protease inhibitor cocktail (EDTA-free) (Roche, cat. # 11836170,001). One gram of glass beads with a diameter of 0.5 mm (YASUI KIKAI, cat. # YGBLA05) was mixed with the cell suspension, and cells were disrupted by vigorous shaking using a beads beater (YASUI KIKAI, MB901). The crude cell extract was collected by brief centrifugation, and glass beads were washed with a small volume of EB (<0.5 mL). The crude extract and washes were combined and clarified by centrifugation twice at 20,000g at 4°C for 10 min. To block endogenous biotin, the cleared lysate was supplemented with 0.2 mg/mL neutralized avidin (FUJIFILM Wako, cat. # 015−24-231), incubated on ice for 30 min, and clarified by centrifugation at 20,000g at 4°C for 10 min. The supernatant was loaded onto a handmade open column packed with 0.5 mL of Strep-Tactin Sepharose (IBA GmbH, cat. # 2-1201-002). After washing with 10 column volumes of EB, bound proteins were eluted with 6 column volumes of EB plus 2.5 mM D-desthiobiotin (IBA GmbH, cat. # 2-1000-025). The protein amounts and compositions of elution fractions were

analyzed by SDS-10% PAGE and stained with FastGene Q-stain (Nippon Genetics, cat. # NE-FG-QS1). Peak fractions were pooled and concentrated using an Amicon Ultra-0.5 centrifugal filter device with a molecular weight cutoff of 30 or 10 kDa (Merck, cat. # UFC503024 or UFC501008). D-desthiobiotin in the peak fraction was diluted more than 100-fold by a repeating dilution with EB and the centrifugal concentration. The concentration of the purified protein was assessed by densitometry using bovine serum albumin (TaKaRa, cat. # T9310A) as a standard following SDS-PAGE and staining with the Q-stain. Purified Ade4-mNG and Ade4ΔIDR-mNG were supplemented with 10-mM DTT and stored on ice.

### In vitro condensation assay

The purified protein was diluted to the indicated concentration in assay buffer (EB plus 5 mM $MgCl_2$) and supplemented with 0%–15% (w/v) PEG 8,000 (FUJIFILM Wako, cat. # 596–09755). In the experiments shown in S5B–S5D Fig, samples were spiked with 10% (w/v) PEG of the indicated molecular size. The mixture was immediately loaded into an assay chamber composed of a glass slide and coverslip attached by double-sided tape. Images were taken after a 2-min incubation. Fluorescent condensation was imaged using the Nikon inverted fluorescent microscope (see above). Images of fluorescent condensation were analyzed by running a Fiji macro (https://github.com/masaktakaine/In-vitro-condensates-quantification). Briefly, the macro automatically detects fluorescent condensation and quantifies its area and fluorescence intensity.

### Quenching of yeast cells for a metabolomics analysis

The quenching of yeast cells before metabolite extraction followed the method reported by Kim and colleagues [69] with slight modifications. Wild-type BY4741 cells were grown to saturation in YPD medium, inoculated into synthetic medium with or without 0.02-mg/mL adenine, and then grown at 30°C for more than 16 h. The preculture was diluted 50–60 times with 10 mL of fresh medium, and cells were grown to ~1.0 × 10$^7$ cells/mL at 30°C. After measuring cell density (cells/mL) with an automated cell counter (TC20, Bio-Rad), 0.9 mL of the cell culture was quenched by a rapid injection into 4.5 mL of pure methanol at -80°C. The suspension was centrifuged at 2,200$g$ at 4°C for 3 min, and the cell pellet was stored at -80°C until extraction. In samples labeled "- ade" and "+ade 24h", cells were cultured for more than 24 h (since the pre-culture) in medium containing 0- or 0.02-mg/mL adenine, respectively. In samples labeled "+ade 1h" and "+ade 7h", adenine was added at a final concentration of 0.02 mg/mL 1 or 7 h before quenching, respectively.

### Metabolite extraction and widely targeted metabolomics profiling

A widely targeted metabolomic analysis was performed as previously described [70]. In brief, each frozen sample in a 1.5-mL plastic tube was homogenized in 200 μL of 50% methanol with glass beads using a microtube homogenizer (TAITEC Corp.) at 41.6 Hz for 2 min. Homogenates were mixed with 400 μL of methanol, 100 μL of $H_2O$, and 200 μL of $CHCl_3$ and vortexed for 20 min. Samples were centrifuged at 20,000$g$ at 4°C for 15 min. The supernatant was mixed with 350 μL of $H_2O$, vortexed for 10 min, and centrifuged at 20,000$g$ at 4°C for 15 min. The aqueous phase was collected, dried in a vacuum concentrator, and re-dissolved in 2-mM ammonium bicarbonate (pH 8.0). Chromatographic separations were performed using an Acquity UPLC H-Class System (Waters) under reverse-phase conditions with an ACQUITY UPLC HSS T3 column (100 × 2.1 mm, particle size of 1.8 μm, Waters) and under HILIC conditions using an ACQUITY UPLC BEH Amide column (100 × 2.1 mm, particle size of 1.7 μm, Waters). Ionized compounds were detected using a Xevo TQD triple quadrupole mass spectrometer coupled to an electrospray ionization source (Waters). The peak areas of target metabolites were analyzed using MassLynx 4.1 software (Waters). Metabolite signals were normalized to the total ion signals of the corresponding sample. To estimate absolute PRPP concentrations, the amount of PRPP in the sample was quantified by comparisons with standard curves obtained for pure PRPP. Intracellular concentrations were then calculated using the previously reported haploid yeast cell volume (42 fL) [71].

## MD simulation

The crystal structure of *E. coli* PPAT (PDB ID: 1ECC) was obtained from the Protein Data Bank and used for all MD simulations [35]. All water molecules and ions were removed before simulations. To build the parameters of ligands ($NH_3$, PRPP, and glutamic acid), their restrained electrostatic potential charges were calculated based on gas-phase HF/6-31(d) quantum mechanics calculations using Gaussian 16. A set of parameters were generated with a general AMBER force field using the antechamber from AMBER tools [72].

The Amber ff14SB force field was employed for all MD simulations [73]. Missing hydrogen atoms were added by LEaP from AMBER tools. Proteins were solvated with the TIP3P water box [74]. The system was neutralized by replacing water with $Na^+$ or $Cl^-$ ions. The particle-mesh Ewald method was employed for the Coulomb interaction [75]. Amber-formatted input files were converted for GROMACS using acpype [76]. All MD simulations were performed using GROMACS 2019.6 [77]. Before the production run, three-step equilibrations were performed for each system. Energy minimization was performed until 10,000 steps. This was followed by 100-ps *NVT* equilibration with the V-rescale thermostat [78] at 300 K under 1 atm. A 100-ps *NPT* equilibration was then conducted with Berendsen coupling [79] at 300 K under 1 atm. A run was performed under the *NPT* condition. The time step of each simulation was 2 fs with the SHAKE [80] and SETTLE [81] algorithms, and a snapshot was recorded every 100 ps during the production run. To examine the results of MD simulations, a Python library MDanalysis was used [82].

## Data analysis

Numerical data were analyzed and plotted using R studio software (R ver. 4.3.1) [83]. Box plots show the 75th and 25th percentiles of data (interquartile range) as the upper and lower edges of the box, respectively, the median as the medial line in the box, and the 1.5× interquartile range as whiskers. The significance of differences between sets of data was indicated by asterisk(s) or *p*-values. The following abbreviations were used unless otherwise noted: $*p < 0.05$; $**p < 0.01$; $***p < 0.001$; NS, not significant ($p > 0.05$). Sparse PLS-DA was performed using the *splsda* function in the mixOmics R package [84]. A one-way ANOVA and subsequent adjustments of *p*-values by the Benjamini–Hochberg method to control the false discovery rate (FDR) were performed using Metaboanalyst 6.0 (https://www.metaboanalyst.ca/MetaboAnalyst/home.xhtml). The top 25 metabolites whose changes strongly correlated with those in the intracellular concentration of adenine were identified using the pattern hunter analysis of Metaboanalyst. All measurements were repeated at least twice to confirm reproducibility.

## Supporting information

**S1 Fig. Additional data regarding the intracellular localization of Ade4.** Related to Fig 1. **(A)** Phase and RFP images of cells expressing Ade4 tagged C-terminally with the red fluorescent protein mKate2. **(B)** A control experiment of time-lapse imaging of the assembly of Ade4-mNG particles by adenine depletion shown in Fig 1E. Cells were grown in medium containing 0.02-mg/mL adenine and washed with medium containing the same concentration of adenine at *t* = 0 min. **(C)** A control experiment of time-lapse imaging of the disassembly of Ade4-mNG particles by adenine supplementation shown in Fig 1F. Cells were grown in the absence of adenine and sterile water was supplemented instead of adenine at *t* = 0 min. Scale bars = 5 μm. **(D–F)** Another example of time-lapse imaging of Ade4-mNG particles, related to Fig 1I–1K. Experimental conditions and graph descriptions are the same as in Fig 1I–1K.
(TIFF)

**S2 Fig. The IDR is conserved in PPATs of various species. (A)** The C-terminal IDR is conserved in PPATs of Ascomycota. Heatmap showing disorder scores for the PPAT amino acid sequences of 81 ascomycota. **(B)** Heatmap showing disorder scores for the PPAT amino acid sequences of 100 organisms, including 19 ascomycota. Phylogenetic groups are color-coded on the left. **(C, D)** Amino acid sequences of IDTs vary across species. Multiple amino acid sequence

alignments of the C-terminal region of PPAT. In (C), three fungi and 13 representative organisms from the following groups are selected: archaea, bacteria, protozoa, plants, nematodes, insects, echinoidea, fish, amphibians, mammals, reptiles, birds, and ascidians. In (D), 13 species are selected from ascomycota. Green boxes indicate the region corresponding to the C-terminal IDR of Ade4. The data for panels **A–B** may be found in S1 Data.
(TIFF)

**S3 Fig. Formation of Ade4 condensates in TORC1-related mutants.** Related to Fig 2. **(A)** Assembly of Ade4-GFP foci in single gene deletion mutants of the upstream regulator and downstream effector of TORC1 signaling. Cells of the indicated genotype expressing Ade4-GFP were grown in medium containing adenine and then incubated in medium without adenine for 45 min before imaging. **(B)** Summary of the percentage of cells showing Ade4-GFP foci in the wild type and 37 single-gene deletion mutants. The percentage of cells with Ade4-GFP foci per field of view was plotted. Data were pooled from 2 to 3 independent experiments. Bars and error bars indicate the mean ± 1 SD. **(C)** The inhibition of ribosome synthesis suppressed the assembly of Ade4-GFP foci. Related to Fig 2J. **(D)** The inhibition of ribosome synthesis suppressed the assembly of Ade4-mNG condensates. Cells expressing Ade4-mNG were grown in medium containing adenine and then incubated in medium without adenine in the presence of 0.05% (v/v) DMSO or 0.05% DMSO plus 5-μg/mL diazaborine for 45 min before imaging. Scale bars = 5 μm. The data for panel **B** may be found in S1 Data.
(TIFF)

**S4 Fig. TORC1 activity and ribosome synthesis are required for the maintenance of Ade4 condensation. (A)** Cells with Ade4-GFP foci in the absence of purine bases were supplemented with 0.02% (v/v) DMSO (DMSO), 0.02% DMSO plus 1-μg/mL rapamycin (Rapa), or 0.02% DMSO plus 5-μg/mL diazaborine (Dia) and then imaged every 15 min for 1 h. Representative cell images are shown. Scale bar = 5 μm. **(B)** Quantification of the data shown in **(A)**. The percentage of cells with foci per one field of view was quantified and plotted. Bars and e rror bars indicate the mean ± 1SD. More than 5 fields of view were imaged for each condition and ~200 cells were examined per field. Note that data at time = 0 were identical across the three groups because untreated cells were imaged and then divided into three subgroups supplemented with each drug. *P*-values were calculated using the two-tailed Tukey–Kramer's multiple comparison test. ***$p < 10^{-4}$. The data for panel **B** may be found in S1 Data.
(TIFF)

**S5 Fig. Purification and basic characterization of the Ade4 protein in vitro. (A)** Purified proteins used in the present study. Proteins were resolved by SDS-PAGE and stained with FastGene Q-stain. **(B)** The in vitro condensation of Ade4-mNG depends on the size of PEG. The supplementation of 0.15-μM Ade4-mNG with 10% PEG of the indicated average molecular weight was performed. **(C, D)** Quantification of data shown in **(B)**. The mean fluorescence intensities of Ade4-mNG particles were box plotted in **(C)**. Cross marks indicate the population mean. The integrated fluorescence intensities of particles were summed per field and plotted in **(D)**. Bars and error bars indicate the mean ± 1 SD. Seven to 8 fields of view were imaged for each condition, and an average of 399 particles per field were examined. *P*-values were calculated using the two-tailed Steel–Dwass multiple comparison test. **(E)** Condensation of Ade4-mNG into particles with crowding agents other than PEG. The supplementation of 0.14-μM Ade4-mNG with the indicated crowding agents was performed. Scale bars = 5 μm. Uncropped gel images are available in S1 Raw images. The data for panels **C** and **D** may be found in S1 Data.
(TIFF)

**S6 Fig. Estimation of the intracellular concentration of Ade4.** Data regarding the protein abundance of the budding yeast Ade4 were downloaded from the Saccharomyces Genome Database (https://www.yeastgenome.org/locus/S000004915/protein). Data were derived from 20 quantitative genome-wide proteomic studies and normalized to a unit of molecules per cell. These values were converted to micromoles assuming that the average cytoplasmic volume of the haploid yeast cell was 42 fL [71]. Data were classified according to the measuring methods (flow cytometry, quantitative

fluorescence microscopy, immunoblotting, and mass spectrometry) and the type of medium (rich or synthetic) and were then plotted. Bars and error bars indicate the mean ± 1 SD. Data may be found in S1 Data. (TIFF)

**S7 Fig. Effects of environmental purine bases on TORC1 activity in yeast cells. (A)** Immunoblot analysis of the phosphorylation state of Rps6. Wild-type (BY4741) cells grown in the absence of purine bases were supplemented with of 0.005% (v/v) DMSO (Rapa: -) or 0.005% DMSO plus 0.5-μg/mL rapamycin (Rapa: +) for 30 min, and total cell extracts were then prepared. Total Rps6 (Rps6) and phosphorylated Rps6 (p-Rps6) were detected as bands at ~30 kDa. The bands of Pgk1 were shown as a loading control. Throughout the figure, asterisks indicate non-specific bands. **(B)** Supplementation with purine bases did not markedly affect the phosphorylation state of Rps6. Wild-type cells grown in the absence of purine bases were supplemented with 20-μg/mL adenine (+Ade) or 25-μg/mL hypoxanthine (+Hyp) for 30 min, and total cell extracts were prepared. The phosphorylation of Rps6 was analyzed by immunoblots as described in **(A)**. **(C)** The removal of purine bases did not markedly affect the phosphorylation state of Rps6. Wild-type cells grown in the presence of 20-μg/mL adenine (+Ade) or 25-μg/mL hypoxanthine (+Hyp) were washed three times with the same medium (mock) or the medium lacking purine bases (-Ade, -Hyp), cultured for 40 min, and total cell extracts were then prepared. The phosphorylation of Rps6 was analyzed by immunoblotting as described in **(A)**. Uncropped images of immunoblots are available in S1 Raw images. (TIFF)

**S8 Fig. Additional data regarding the metabolomics analysis.** Related to Fig 4. **(A)** Changes in the indicated metabolites. Experimental conditions and graph descriptions are the same as in Fig 4D. Data may be found in S3 Table. **(B)** The intracellular concentration of PRPP decreases in the presence of environmental adenine. The amount of PRPP in each sample was converted to an intracellular concentration and plotted. *P*-values were calculated using the two-tailed Tukey–Kramer's multiple comparison test. The data for panel **B** may be found in S1 Data. (TIFF)

**S9 Fig. Alanosine inhibits the dissolution of Ade4 foci in the presence of hypoxanthine.** Related to Fig 5. **(A)** Alanosine inhibits the adenylosuccinate synthetase Ade12. **(B)** Assembly of Ade4-GFP foci in the presence of alanosine. Cells expressing Ade4-GFP were incubated in medium lacking adenine in the presence of 0 (mock) or 30 μg/mL alanosine (+alanosine) for 40 min to allow for the assembly of Ade4-GFP foci and were then imaged (SC). Cells were further treated with the same medium supplemented with 24-μg/mL hypoxanthine for 15 min and were then imaged again (SC + hyp). Scale bar = 5 μm. **(C)** Quantification of data shown in **(B)**. The percentage of cells with foci per field of view (containing 182 ± 46 cells) was quantified and plotted. Data were pooled from two independent experiments. Bars and error bars indicate the mean ± 1 SD. The significance of differences was tested using the unpaired two-tailed Welch's *t* test. *P*-values vs. the mock are shown. The data for panel **C** may be found in S1 Data. (TIFF)

**S10 Fig. Additional data regarding the assembly of Ade4 condensates in vitro, related to Fig 6. (A, B)** Effects of purine base derivatives on the in vitro assembly of Ade4 condensates. The supplementation of 0.2-μM Ade4-mNG with 10% PEG8K was performed in the presence of the indicated compound. Glucose (Glu) was used as a negative control. The integrated fluorescence intensities of condensates were summed per field and plotted in **(A)**. The mean fluorescence intensities of Ade4-mNG condensates were averaged per field and plotted in **(B)**. Bars and error bars indicate the mean ± 1 SD. Twenty fields of view were imaged for each condition from two independent preparations, and an average of 1,452 particles per field were examined. *P*-values were calculated using the two-tailed Steel's multiple comparison test. Glu, glucose; Ade, adenine; Ino, inosine; d-Ino, deoxy-inosine. **(C)** Effects of purine nucleotides on the in vitro assembly of Ade4 condensates. Data were from the same experiment shown in Fig 6A. The mean fluorescence intensities of Ade4-mNG

 

condensates were averaged per field and plotted. Bars and error bars indicate the mean ± 1 SD. *P*-values were calculated using the two-tailed Tukey–Kramer's multiple comparison test. **(D–F)** The supplementation of 0.2-µM Ade4-mNG with 10% PEG8K was performed in the presence of the indicated concentrations of ADP **(D)**, ATP **(E)**, and GMP **(F)**. The integrated fluorescence intensities of the condensates were summed per field, normalized to the mean at 0 µM, and plotted. Bars and error bars indicate the mean ± 1 SD. From two independent preparations, 25–31 fields of view were imaged for each condition, and an average of 1,200 particles per field were examined. *P*-values were calculated using the two-tailed Steel's multiple comparison test. **(G, H)** Inhibitory effect of SAH on the in vitro condensation of Ade4-mNG. The supplementation of 0.2-µM Ade4-mNG with 10% PEG8K was performed in the presence of 0.5% (v/v) DMSO and 500-µM SAH or AMP and imaged. The average fluorescence intensities **(G)** and integrated fluorescence intensities **(H)** of the particles summed per field were normalized to the mean of DMSO alone and plotted. Bars and error bars indicate the mean ± 1 SD. From 4 independent preparations, 53–57 fields of view were imaged for each condition. *P*-values were calculated using the two-tailed Dunnett's multiple comparison test. The data for panels **A–H** may be found in S1 Data.
(TIFF)

**S11 Fig. Effects of IMP on the assembly of Ade4 condensates in vitro. (A, B)** IMP did not affect the in vitro condensation of Ade4-mNG. The supplementation of 0.2-µM Ade4-mNG with 10% PEG8K was performed in the presence of 500 µM IMP or AMP. The average fluorescence intensities **(A)** and integrated fluorescence intensities **(B)** of the condensates summed per field were normalized to the mean of the mock and plotted. Bars and error bars indicate the mean ± 1 SD. From 4 independent preparations, 62–72 fields of view were imaged for each condition. *P*-values were calculated using the two-tailed Steel's multiple comparison test. The data for panels **A** and **B** may be found in S1 Data.
(TIFF)

**S12 Fig. PRPP facilitates the assembly of Ade4 condensates, related to** Fig 7**. (A)** PRPP augments the in vitro assembly of Ade4 condensates. Data were from the same experiment shown in Fig 7A. The mean fluorescence intensities of Ade4-mNG condensates were averaged per field and plotted. Throughout the figure, bars and error bars indicate the mean ± 1 SD. *P*-values were calculated using the two-tailed Steel's multiple comparison test. **(B, C)** PRPP promotes the formation of Ade4 condensates in vitro at suboptimal PEG concentrations. Data were from the same experiment shown in Fig 7B and 7C. The mean fluorescence intensities of Ade4-mNG condensates were averaged per field and plotted in **(B)**. ND, not determined. Representative images at 5% PEG are shown in **(C)**. **(D)** PRPP antagonizes the inhibitory effects of AMP. Data were from the same experiment shown in Fig 7D and 7E. The mean fluorescence intensities of Ade4-mNG condensates were averaged per field, normalized to the mean of the mock control (no AMP and no PRPP), and plotted. *P*-values were calculated using the two-tailed Steel's multiple comparison test. **(E–G)** In the presence of 1-mM AMP and the indicated concentrations of PRPP, 0.2-µM Ade4-mNG was supplemented with 5% or 7.5% PEG8K. From 2 to 3 independent preparations, 24–36 fields of view were imaged for each condition. **(E)** The mean fluorescence intensities of Ade4-mNG condensates were averaged per field and plotted. Throughout the figure, bars and error bars indicate the mean ± 1 SD. **(F)** The integrated fluorescence intensities of the condensates were summed per field and plotted. *P*-values were calculated using the two-tailed Mann–Whitney U test. Representative images under 5% PEG conditions are shown in **(G)**. Scale bars = 5 µm. Note that although PRPP increased the number of extremely small condensates, it failed to strongly promote the condensation of Ade4-mNG. The data for panels **A**, **B**, and **D–F** may be found in S1 Data.
(TIFF)

**S13 Fig. Characterization of the Ade4-D373A/D374 protein in vitro, related to** Fig 7**. (A)** The amino acid residues of the PRPP binding motif in PPAT are well conserved among species. **(B)** Interactions between an AMP and specific amino acid residues in the PRPP binding motif in *E. coli* PPAT (pdb file: 1ecj). The dotted line indicates a hydrogen bond. **(C)** The purified Ade4-D373A/D374A-mNG (Ade4-DA-mNG) protein. Proteins were resolved by SDS-PAGE and stained with

FastGene Q-stain. **(D)** Ade4-DA-mNG condensated into particles under molecular crowding conditions. The supplementation of 0.2-µM Ade4-mNG (WT) and Ade4-DA-mNG (DA) with 10% PEG8K was performed and imaged. The integrated fluorescence intensities of the condensates were summed per field and plotted. From 2 independent preparations, >36 fields of view were imaged for each condition. *P*-values were calculated using the two-tailed Mann–Whitney U test. **(E, F)** PRPP did not antagonize AMP in condensate formation in the DA mutant. Data were from the same experiment shown in Fig 7K. The mean fluorescence intensities of Ade4-D373A/D374A-mNG condensates were averaged per field, normalized to the mean of the mock control (no AMP or PRPP), and plotted in **(E)**. *P*-values were calculated using the two-tailed Steel's multiple comparison test. Representative images at 7.5% PEG are shown in **(F)**. Scale bars = 5 µm. Uncropped gel images are available in S1 Raw images. The data for panels **D** and **E** may be found in S1 Data.
(TIFF)

**S14 Fig. Possible enzymatic reaction pathways of PPAT. (A)** Intramolecular channeling of glutamine-derived NH3 proposed by Krahn and colleagues [35]. **(B)** Mechanism of $NH_3$-dependent activity proposed by Chen and colleagues [54]. External $NH_3$ enters the channel from the glutamine site irrespective of glutamine binding. **(C)** The MD simulation in the present study suggests that glutamine-derived $NH_3$ leaks through the front channel following the attack of exogenous $NH_3$. **(D)** Condensation-dependent activation of Ade4 proposed by the present study. Glutamine-derived $NH_3$ leaked from the front channel rapidly diffuses away. In the Ade4 condensate, the opportunity for $NH_3$ to be captured by other molecules and further reacted with PRPP will increase. This activation may be described as intermolecular $NH_3$ channeling. PRPP is not depicted in the schema.
(TIFF)

**S15 Fig. The formation of Ade4 condensates is important for efficient DPS during cell growth, related to** Fig 9. **(A, B)** Growth of cells expressing Ade4-mGFP, Ade4-GFP, Ade4ΔIDR-mGFP, and Ade4ΔIDR-GFP on solid media. Cells of each strain were serially diluted (5-fold), spotted on 2% glucose medium containing 38 mM AS **(A)** and 8 mM proline (no AS) **(B)**, and then grown at 30°C for 2 days. **(C)** Growth curves of cells expressing Ade4-mGFP, Ade4-GFP, Ade4ΔIDR-mGFP, and Ade4ΔIDR-GFP in liquid media without adenine. Cells were grown at 30°C for 25 h in medium containing the indicated nitrogen source in the absence of adenine. Data are the mean of 10–15 replicates, and the shaded area indicates ±1 SD. **(D)** Quantification of the data shown in **(C)**. The time to half-maximum density (corresponding to normalized $OD_{595}$ = 0.5) was calculated and plotted. Bars and error bars indicate the mean ± 1 SD. *P*-values were calculated using the Tukey–Kramer multiple comparison test. **(E)** Immunoblot analysis of the protein level of Ade4 tagged C-terminally with GFP or mGFP. Cells expressing the indicated Ade4 construct were grown to the mid-log phase in the absence of adenine, supplemented with 0.02-mg/mL adenine, cultured for another 20 min to dissolve Ade4 condensation, and total cell extracts were then prepared. Ade4 proteins were detected as bands at ~85 kDa by an anti-GFP antibody. Asterisks indicate non-specific bands. The bands of PGK1 were shown as a loading control. Uncropped images of immunoblots are available in S1 Raw images. The data for panels **C** and **D** may be found in S1 Data.
(TIFF)

**S16 Fig. Summary of conditions for Ade4 condensation.** In vivo results are based on data shown in Figs 1, 2, and 4. In vitro results are based on data shown in Figs 3C, 6C, 7A–B, 7D, and S12F.
(TIFF)

**S1 Table. Metabolomics profile of budding yeast cells grown in the absence and presence of adenine.** The normalized peak area is the signal peak area corrected by the total ion chromatogram. Each group contains seven biological replicates. All data of the normalized peak area were subjected to sparse PLS-DA shown in Fig 4B. Means and standard deviations (SD) across replicates are shown as a reference. *P*-values of the one-way ANOVA and FDR-adjusted *p*-values were calculated using MetaboAnalyst 6.0. Blue bold values indicate less than 0.05.
(XLSX)

**S2 Table. Metabolic profile of 56 metabolites whose concentrations were significantly changed by adenine.** Each group contains seven biological replicates. Values are the normalized peak area further converted to Z-scores across samples. Data were subjected to the heatmap analysis shown in Fig 4C.
(XLSX)

**S3 Table. Changes in purine nucleotides and their derivatives.** Each group contains seven biological replicates. Data were converted to FC relative to the mean at 0 h (in the absence of adenine) and were subjected to the analysis shown in Figs 4D and S8A.
(XLSX)

**S4 Table. Strains used in this study.**
(XLSX)

**S5 Table. Plasmids used in this study.**
(XLSX)

**S6 Table. Primers used in this study.**
(XLSX)

**S1 Raw Images. Uncropped images of immunoblots and gel.** Each figure panel displays uncropped, minimally adjusted images of immunoblots or stained gels with annotations of experimental samples and molecular weights. The cropped area is indicated by a dotted rectangle. Lanes not included in the final figure are marked with an "X" above the lane.
(PDF)

**S1 Data. Raw data set.**
(XLSX)

**S1 Movie. Assembly of Ade4-mNG condensates by adenine depletion.** Time-lapse imaging of the assembly of Ade4-mNG condensates by adenine depletion. Cells were grown in medium containing 0.02-mg/mL adenine and washed with medium lacking adenine at $t = 0$ min. The three cells in the upper right are shown in Fig 1E. Scale bar = 5 μm.
(AVI)

**S2 Movie. Assembly of Ade4-mNG condensates (another example).** Another example of cells showing the assembly of Ade4-mNG condensates by adenine depletion. Experimental conditions are the same as in S1 Movie. Scale bar = 5 μm.
(AVI)

**S3 Movie. Assembly of Ade4-mNG condensates (mock adenine depletion).** A control experiment of time-lapse imaging of the assembly of Ade4-mNG condensates by adenine depletion. Cells were grown in medium containing 0.02-mg/mL adenine and washed with medium containing the same concentration of adenine at $t = 0$ min. The rightmost cell is shown in S1B Fig. Scale bar = 5 μm.
(AVI)

**S4 Movie. Disassembly of Ade4-mNG condensates by adenine supplementation.** Time-lapse imaging of the disassembly of Ade4-mNG condensates by adenine supplementation. Cells were grown in the absence of adenine and 0.02-mg/mL adenine was supplemented at $t = 0$ min. The four cells in the upper left are shown in Fig 1F. Scale bar = 5 μm.
(AVI)

**S5 Movie. Disassembly of Ade4-mNG condensates (mock adenine supplementation).** A control experiment of time-lapse imaging of the disassembly of Ade4-mNG condensates by adenine supplementation. Cells were grown in the

absence of adenine and sterile water was supplemented in place of adenine at $t = 0$ min. The two cells in the center are shown in S1C Fig. Scale bar = 5 μm.
(AVI)

**S6 Movie. Disassembly of Ade4-mNG condensates (another example of mock adenine supplementation).** Another example of cells showing Ade4-mNG condensates after a mock treatment with adenine. Experimental conditions are the same as in S5 Movie. Scale bar = 5 μm.
(AVI)

**S7 Movie. Motility of Ade4-mNG condensates.** A close examination of the motility of Ade4-mNG condensates. Another example of data shown in Fig 1I. Cells bearing condensates were imaged on a single z-plane at 1-s intervals. The intensity of mNG is represented by an inverted grayscale. Scale bar = 5 μm.
(AVI)

**S8 Movie. MD simulation from the semi-holo state.** Time-lapse video of the MD simulation of $NH_3$ and PPAT molecules in the semi-holo state, related to Fig 8E. This movie was created from MD snapshots during the simulation. The state of the Arg gate is also indicated. In this example, $NH_3$ was eventually released through the front channel (indicated by a circle).
(MOV)

## Acknowledgments

The authors thank the IMCR-JURSC in Gunma University for sharing research equipment, and Dr K. Ohashi for technical suggestions and sharing reagents for the immunoblot analysis of Rps6. This study used the computational resources of Cygnus provided by the Multidisciplinary Cooperative Research Program at the Center for Computational Sciences (Project Code: MOLBIO), University of Tsukuba.

## Author contributions

**Conceptualization:** Masak Takaine.

**Data curation:** Masak Takaine, Rikuri Morita, Yuto Yoshinari, Takashi Nishimura.

**Funding acquisition:** Masak Takaine.

**Investigation:** Masak Takaine, Rikuri Morita, Yuto Yoshinari, Takashi Nishimura.

**Methodology:** Masak Takaine, Takashi Nishimura.

**Project administration:** Masak Takaine.

**Resources:** Masak Takaine.

**Supervision:** Masak Takaine.

**Writing – original draft:** Masak Takaine, Rikuri Morita.

**Writing – review & editing:** Masak Takaine, Rikuri Morita, Yuto Yoshinari, Takashi Nishimura.

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
