## [Editor Report · Decision Letter 0]

20 Aug 2024

Dear Dr Takaine, 

Thank you for submitting your manuscript entitled "Condensation of PRPP amidotransferase by phase separation promotes de novo purine synthesis through intermolecular intermediate channeling in yeast" for consideration as a Research Article by PLOS Biology.

Your manuscript has now been evaluated by the PLOS Biology editorial staff as well as by an academic editor with relevant expertise and I am writing to let you know that we would like to send your submission out for external peer review.

Once your full submission is complete, your paper will undergo a series of checks in preparation for peer review. After your manuscript has passed the checks it will be sent out for review. To provide the metadata for your submission, please Login to Editorial Manager (https://www.editorialmanager.com/pbiology) within two working days, i.e. by Aug 22 2024 11:59PM.

Kind regards,

Ines

--

Ines Alvarez-Garcia, PhD

Senior Editor

PLOS Biology

---

## [Decision Letter · Decision Letter 1]

26 Nov 2024

Dear Dr Takaine,

Thanks again for your patience while your manuscript entitled "Condensation of PRPP amidotransferase by phase separation promotes de novo purine synthesis through intermolecular intermediate channeling in yeast" was peer-reviewed at PLOS Biology. The manuscript has now been evaluated by the PLOS Biology editors, an Academic Editor with relevant expertise, and by one reviewer. We were expecting to receive an additional report, but the reviewer never sent it. Thus, the Academic Editor has also provided detailed comments on your manuscript.

The review is attached below along with the Academic Editor's comments. As you will see, the reviewer finds your conclusions interesting and worth pursuing for publication, however the reviewer also raises several issues that would need to be addressed. These include analysing how quickly Ade4 condensates disassemble upon Rapamycin treatment, and whether or not this can be prevented with an inhibitor of polysome collapse. This reviewer also thinks you should explore if the PEG concentration correlates with levels of intracellular crowding. The Academic Editor agrees with these points, but has also added a couple of requests that would need to be addressed.

In light of the reviews, which you will find at the end of this email, we would like to invite you to revise the work to thoroughly address the reviewers' reports. Given the extent of revision needed, we cannot make a decision about publication until we have seen the revised manuscript and your response to the reviewers' comments. Your revised manuscript is likely to be sent for further evaluation by the reviewer.

**IMPORTANT - SUBMITTING YOUR REVISION**

3. Resubmission Checklist

a) *PLOS Data Policy*

b) *Published Peer Review*

d) *Blurb*

Please also provide a blurb which (if accepted) will be included in our weekly and monthly Electronic Table of Contents, sent out to readers of PLOS Biology, and may be used to promote your article in social media. The blurb should be about 30-40 words long and is subject to editorial changes. It should, without exaggeration, entice people to read your manuscript. It should not be redundant with the title and should not contain acronyms or abbreviations. For examples, view our author guidelines: https://journals.plos.org/plosbiology/s/revising-your-manuscript#loc-blurb

Sincerely,

Ines

--

Ines Alvarez-Garcia, PhD

Senior Editor

PLOS Biology

Reviewers' comments

Rev. 1:

In this interesting study, Takaine et al investigate how the S. cerevisiae PRPP amidotransferase Ade4 is regulated in different nutrient conditions. They describe that Ade4 forms highly dynamic and reversible foci upon purine starvation. Formation of these foci can be prevented by deletion of a conserved IDR region in the protein or by addition of hexane diol, leading the authors to conclude that the observed foci represent liquid condensates. These condensates seem to depend on high levels of intracellular crowding as Ade4 focus formation depends on TORC1 induced ribosome biogenesis and synthetic crowding agents (PEG, Ficoll) induce purified Ade4 to undergo a phase transition in vitro. Furthermore, the authors describe how metabolic changes that occur in response to purine starvation promote Ade4 condensation. In vitro, Ade4 condensation is strongly increased by binding to its substrate PRPP and slightly inhibited by AMP. At high crowding levels the positive effect of PRPP is however only modest. To assess the functional relevance of the identified metabolite- and crowding dependent condensation of Ade4, the authors performed molecular dynamics simulations that suggest that condensation supports substrate channeling between the active sites of Ade4. I am not qualified to assess the quality of these molecular dynamics calculations. Finally, the authors show that deletion of the IDR domain in Ade4 reduces cell fitness in purine depleted conditions.

Overall, this is an exciting manuscript with an impressive amount data. The experimental data are of high quality and generally support the author's conclusions. I have a number of points the authors might want to address before publication

1. From the observation that Ade4 focus formation depends on TORC1 and Sfp1 activity, the authors conclude that ribosome biogenesis dependent crowding is needed for condensate formation. A recent study by the Holt lab (PMID: 39059370 ) suggests that polysomes rather than individual ribosomes play a major role in cytoplasmic crowding. As polysomes collapse much faster than overall ribosome concentration decreases, it would be interesting to see how quickly Ade4 condensates disassemble upon Rapamycin treatment and whether it can be prevented by simultaneous treatment with Cycloheximide (prevents polysome collapse).

2. While PRPP can boost condensate formation at intermediate PEG concentrations, at high concentrations (10%) PRPP is much less potent at inducing Ade4 condensation. Whether or not PRPP levels are important in regulating Ade4 condensates therefore depends on the level of crowding in the cytoplasm. It would be useful if the authors could relate the PEG concentration to levels of intracellular crowding.

3. It would help if the negative control for the in vitro condensation experiments (mNG, Figure S4B) would be shown next to the Ade4-mNG condensates in the main figure and displayed in the same way (inverted intensities).

4. The authors show that ADE4-IDR∆ mutants display a growth defect on ADE- which is accentuated in the absence of ammonium. This is consistent with the idea that Ade4 condensation promotes purine synthesis by substrate channeling. Another possible interpretation of these data is that deleting the IDR domain reduces the intrinsic enzymatic activity of Ade4 and that the observed growth defect on Ade- is caused by this mutation. Do the authors have data to exclude this possibility. If not, they should at least discuss this possibility.

Comments from the Academic Editor

I share reviewer 1’s enthusiasm for this manuscript and would like to see the authors address their comments. I additionally have a couple of requests:

i/ Line 245: …However, environmental purine bases, including adenine, do not affect TORC1 activity in mammalian cells 29, 30. I don’t think these references come to this conclusion. I would recommend determining if TORC1 activity is affected by the treatments performed in figure 5B. This may be particularly important given that phosphorylation of Prs5 is potentially regulated downstream of TORC1 (DOI: 10.1101/gad.532109).

ii/ The molecular dynamics data are quite compelling, but I would like to see these hypotheses confirmed in vivo if possible. For example, does the Arg333Ala mutant have any phenotypes in yeast? Could it perhaps rescue the slow growth phenotype when introduced into the Ade4ΔIDR-3HA gene?

---

## [Decision Letter · Decision Letter 2]

12 Mar 2025

Dear Dr Takaine,

Thank you for the submission of your revised Research Article entitled "Phase separation of the PRPP amidotransferase into dynamic condensates promotes de novo purine synthesis in yeast" for publication in PLOS Biology. On behalf of my colleagues and the Academic Editor, Robbie Loewith, I am delighted to let you know that we can in principle accept your manuscript for publication, provided you address any remaining formatting and reporting issues. These will be detailed in an email you should receive within 2-3 business days from our colleagues in the journal operations team; no action is required from you until then. Please note that we will not be able to formally accept your manuscript and schedule it for publication until you have completed any requested changes.

PRESS

Sincerely, 

Ines

--

Ines Alvarez-Garcia, PhD

Senior Editor

PLOS Biology
